# 3D Printing of Hydrogel Polysaccharides for Biomedical Applications: A Review

**DOI:** 10.3390/biomedicines13030731

**Published:** 2025-03-17

**Authors:** Mohammad Aghajani, Hamid Reza Garshasbi, Seyed Morteza Naghib, M. R. Mozafari

**Affiliations:** 1Nanotechnology Department, School of Advanced Technologies, Iran University of Science and Technology (IUST), Tehran 1684613114, Iran; soroshaghajani47@yahoo.com (M.A.);; 2Australasian Nanoscience and Nanotechnology Initiative (ANNI), Monash University LPO, Clayton, Melbourne, VIC 3168, Australia

**Keywords:** 3D printing, polysaccharides, tissue engineering, hydrogels, bioinks

## Abstract

Additive manufacturing, also known as 3D printing, is becoming more and more popular because of its wide range of materials and flexibility in design. Layer by layer, 3D complex structures can be generated by the revolutionary computer-aided process known as 3D bioprinting. It is particularly crucial for youngsters and elderly patients and is a useful tool for tailored pharmaceutical therapy. A lot of research has been carried out recently on the use of polysaccharides as matrices for tissue engineering and medication delivery. Still, there is a great need to create affordable, sustainable bioink materials with high-quality mechanical, viscoelastic, and thermal properties as well as biocompatibility and biodegradability. The primary biological substances (biopolymers) chosen for the bioink formulation are proteins and polysaccharides, among the several resources utilized for the creation of such structures. These naturally occurring biomaterials give macromolecular structure and mechanical qualities (biomimicry), are generally compatible with tissues and cells (biocompatibility), and are harmonious with biological digesting processes (biodegradability). However, the primary difficulty with the cell-laden printing technique (bioprinting) is the rheological characteristics of these natural-based bioinks. Polysaccharides are widely used because they are abundant and reasonably priced natural polymers. Additionally, they serve as excipients in formulations for pharmaceuticals, nutraceuticals, and cosmetics. The remarkable benefits of biological polysaccharides—biocompatibility, biodegradability, safety, non-immunogenicity, and absence of secondary pollution—make them ideal 3D printing substrates. The purpose of this publication is to examine recent developments and challenges related to the 3D printing of stimuli-responsive polysaccharides for site-specific medication administration and tissue engineering.

## 1. Introduction

One can print a sequence of materials layer by layer using additive manufacturing (AM), also known as 3D printing, with the ability to modify the characteristics and form of each layer. A 3D printer often produces a substantial, intricate, and personalized structure that has already been created as an image in a digital brain. In a nutshell, AM can be divided into five major categories: stereolithography (SLA), extrusion-based printing, binder jetting, selective laser sintering (SLS), and inject printing [1,2]. Comparing this current technology to traditional processing methods reveals a number of advantages. For example, AM may easily modify the final products, customize product series, regulate the fabrication process appropriately, and produce highly reproducible results. The capacity to create various shapes (bone, ear, nose, and meniscus) with excessive porosity is especially important in the realm of biomedical engineering. For example, 3D-printed scaffolds’ porous shape makes it easier for nutrients to reach the cells, encourages cell survival, and gives the cell the right medium for organ or tissue regeneration [3,4]. The most notable uses of 3D printing technology include the creation of organs, accurate drug printing, medical phantoms, and several facets of cancer treatment, from medication administration to diagnostics. In fact, the finished 3D-printed structure does not exhibit the dynamic pattern of form change, swelling, self-assembly, self-repairing, multifunctionality, and shape-shifting characteristics over time. However, a lack of dynamism undermines and adversely impacts biomimicry [5,6].

The soft biomaterials called bioinks, which are made of living cells and/or biologically active chemicals, are essential to the successful creation of 3D structures. Bioinks must have a number of essential qualities, such as printability (i.e., viscoelastic properties), printing fidelity, mechanical integrity, and biocompatibility, because bioprinting is largely dependent on the deposition of bioinks into designed shapes created using computer-aided design (CAD) models [7,8,9]. Using physiologically active compounds and/or living cells integrated into the bioink is known as 3D bioprinting.

The biomedical applications of polysaccharides, primarily chitosan, alginate, agarose, starch, glycogen, and cellulose, as well as their mixes, are numerous and include imaging, diagnostic, therapeutic, delivery, and biosensing applications [10,11,12,13,14,15,16,17,18]. Polysaccharides’ bioactivity explains why they are used to treat conditions including immunoregulatory, antiviral, and tumor conditions [19,20]. Despite being well-known for their suitable biocompatibility and harmless nature, polysaccharides have subpar mechanical qualities. The difficulty of extracting and purifying polysaccharides may be another drawback. Additionally, some investigations highlight the poor stability of polysaccharide-based scaffolds in biological media, which calls for adjustments to polysaccharide extraction and processing conditions. Therefore, new technologies such as electrospinning were required to manufacture predesigned biomaterials based on polysaccharides [21,22]

As a relatively new field of innovation, tissue engineering (TE) is the product of a complicated combination of pre-existing lines of study from three distinct fields of knowledge: clinical medicine, engineering, and the biomedical sciences. Clinical difficulties relating to the immunological reaction to allograft transplants and the lack of tissue samples for replacing a damaged area following an accident or injury give rise to the TE area. Furthermore, autografts and allograft-based therapy techniques are linked to high morbidity, limited availability, and the need for immunosuppressive medications for long-term care [23,24].

In order to help heal wounds or replace failing organs, TE seeks to preserve, repair, enhance, or replace various tissue types. Eventually, the in vitro-formed tissue will be included or integrated into the body. TE suggests creating temporary structures with structural and functional characteristics that resemble the original tissue in order to achieve these objectives. Various bioengineered structures have been created in this sector using a variety of methods, including bioprinting, injectable hydrogels, and matrix decellularization [24,25,26,27,28]. The initial objective of TE was to create tissue constructs through biological or engineering methods; however, advancements in this field have been assisting in the investigation of more comprehensive approaches that incorporate the utilization of a biomaterial or biomimetic platform in conjunction with cells and biological elements that stimulate the process of tissue regeneration [25].

In order to create these biomimetic 3D supports, bioprinting is a promising technology that has transformed a number of disciplines, including regenerative medicine and TE. By combining various biomaterials with live and physiologically relevant cells, this approach allows for the high-throughput production of tissue models by forming porous structures with a regulated architecture. In addition to facilitating proper cell distribution, adhesion, and growth inside the biomaterial scaffold, bioprinting also improves gas and nutrient exchange and cell–cell communication because of the vast surface area and linked holes [29,30]. By providing 3D scaffolds with spatial depth and more realistic cell–cell communication, bioprinting can help overcome the absence of a dynamic and complex tissue structure in 2D cell culture and improve our understanding and simulation of in vivo physiology. Additionally, the success of bioprinting technology depends on the effectiveness of 3D bioprinters and the availability of bioinks (polymer solutions and live cells) with appropriate structural, physicochemical, mechanical, thermal, and biological properties [30].

Bioinks must possess the appropriate mechanical, rheological, chemical, and biological properties in order to be deemed appropriate for bioprinting [31]. This means that the materials used to make the bioink formulation must be able to be printed, have the right mechanical properties for the tissue that is being targeted, have high resolution and shape fidelity if necessary, and be biocompatible, biodegradable, bioactive, and reliable in order to mimic the microenvironment of tissues [32]. Therefore, the selection of (bio)printable materials remains the bottleneck in bioprinting technology because the bioprinting technique and cell inclusion, respectively, necessitate the use of a polymer solution with appropriate rheological and biological properties.

Since bioprinter-specific criteria must be taken into account, it is especially important to ensure biocompatibility, biodegradability, and unique structural and mechanical features when producing a hydrogel formulation for TE. Additive manufacturing can be used to create a wide range of drug delivery methods and formulations, including transdermal patches, vaginal and rectal delivery systems, drug-loaded implants, and surgical patches, in addition to various oral solid dosage forms [33]. This restriction may result from regulatory constraints and health and safety concerns, but it may also be related to production speed limitations, which can be slower than with traditional manufacturing techniques [34].

Despite a growing number of intriguing reviews on medicine delivery and additive manufacturing, there remains a lack of knowledge about printing polysaccharides. With the goal of highlighting promising avenues, we will examine current developments in the 3D printing of polysaccharides with an emphasis on tissue engineering and localized drug delivery applications in this paper.

## 2. 3D Printing Techniques

3D printing has improved as a technique, using materials like polymers, metals, and ceramics to manufacture items layer by layer with various shapes [35]. Many 3D printing methods are presently accessible for many uses. The methods in question may be categorized into several distinct areas: binder jetting, sheet lamination, powder bed fusion, and material extrusion (Figure 1) [36].

The production of materials for curing or preventing diseases in the pharmaceutical and medical fields does not always require all the procedures, notwithstanding their diversity. Most recent evaluations were thoroughly discussed [38,39,40]. The most often utilized 3D printing methods in the medical industry are fused deposition modeling (FDM), extrusion-based bioprinting, inkjet, and PolyJet, which are included in Table 1.

The automotive and aerospace sectors have used 3D printing technology for over three decades. This technique was only used to print anatomical prototypes for educational instruction. Three-dimensional printing in the healthcare and pharmaceutical sectors has increased significantly due to recent developments in producing new biodegradable components. Three-dimensional printing technology is widely used and is quickly gaining ground in the medical sector. The bioprinting of organs and live scaffolds in regenerative medicine, the customization of implants and prostheses, and the development of surgical instruments and patient-specific biomedical models have all resulted in significant changes to the healthcare system.

### 2.1. Fused Deposition Modeling (FDM) or Free Form Fabrication (FFF)

Fused deposition modeling is a 3D printing technology that uses a continuous filament of thermoplastic material for the fabrication of complete structures (Figure 1A) [52,53]. FDM’s main advantage is that it is able to rapidly and affordably prepare a large number of shapes at cheaper prices than traditional production methods [52]. In the late 1980s, S. Scott Crump designed FDM, which Stratasys Inc., based in Edina, Minnesota, USA, started to market in the early 1990s [54]. A moving printhead in the X and Y directions, a movable platform in the Z direction, and a base material in the form of a cylindrical filament typically make up its setup. Based on the computer-aided design (CAD) design, the printhead is made up of a nozzle that is heated where the polymer is melted or softened before being laid down as a layer onto the platform, where it cools and solidifies [55]. A computer manages all of the head’s movements and the base materials as the subsequent layers are deposited on top of the previously printed layers and fuse with them to create the thing from the bottom up. The layer thickness, shell, and infill pattern are the three main factors influencing the quality of FDM 3D-printed scaffolds. When assessing an FDM-printed product, two of the most crucial characteristics are dimensional accuracy and surface roughness. In terms of dimensional accuracy, the general shrinking behavior of FDM components is determined by the glass transition temperature of the filament’s material, whereas improved dimensional precision is the consequence of the extruded material immediately solidifying [56]. The majority of filament substances used in FDM printing are thermoplastics, or materials that melt at extreme temperatures and immediately solidify at low ones. For pharmaceutical and medical applications, FDM is frequently used to print PLA; however, other materials, like cellulose derivatives or Eudragit^®^, have been investigated in recent years [57,58].

Hot melt extrusion (HME) is typically used to create polymer filaments appropriate for FDM. Since HME was first discovered in the early 1930s, it was used to manufacture rubber and plastic products in a variety of forms, including films, fibers, and granules [59]. The technique is relatively easy: a homogeneous filament is generated by pumping melted polymeric material through revolving screws via an extruder barrel. As anticipated, the printing process is significantly impacted by the extruded filament’s characteristics. A constant diameter along the filament’s length guarantees consistent printing. Additionally, the filaments require a suitable ratio between flexibility and hardness to enable ongoing 3D printing. In fact, soft filaments clog the FDM printer, while fragile filaments shatter during printing [60].

Uses for FDM vary and include the automotive, aerospace, industrial, and medical industries. HME first creates the proper drug filaments for printing drug delivery systems, that involve the drug disseminated in the polymeric matrix and uniformly distributed during extrusion. Another technique for recharging filaments is to soak them in a pharmacological solution. The completed drug-loaded object is then printed by inserting the drug-containing filament into an FDM printer [56]. The absence of solvents is one of FDM’s advantages, but its temperature is a drawback. Since the polymeric material needs to be melted, FDM does, in fact, require high temperatures, which increases the possibility of degradation for the active pharmaceutical ingredients (APIs) that are integrated [54].

While research is being conducted to develop systems that require lower temperatures, numerous research organizations are examining the suitability of FDM in drug delivery systems for less heat-sensitive medicines.

### 2.2. Laminated Object Manufacturing

Another additive manufacturing method used in bioprinting to create three-dimensional tissue structures is laminated object manufacture (LOM). The idea behind LOM is to create the desired object by layer-by-layer assembling tiny sheets of material, usually paper or polymer, that are joined by heat or adhesive. Biocompatible materials, such as cellulose-based paper or biodegradable polymers, are frequently utilized as scaffold construction building blocks in bioprinting applications [61]. A computer-aided design (CAD) model is first digitally sliced, and then the material sheets are cut or laser-cut to fit the contour of each layer. The whole 3D structure is subsequently produced by stacking and bonding these layers. High build speed, cheap cost, and the capacity to create large-scale structures are some benefits of LOM. Enhancing cell integration and survival, investigating new materials for tissue engineering applications, and fine-tuning process parameters to gain exact control over scaffold features are the main goals of current LOM-based bioprinting research. Recent research has looked into creating scaffolds with controlled porosity and hierarchical topologies utilizing LOM in order to replicate the natural tissue environment [62]. Furthermore, multi-material printing and integration with sacrificial materials, two developments in LOM technology, hold promise for creating intricate, useful tissue structures for drug discovery and regenerative medicine [63].

### 2.3. Selective Deposition Laminations

An additive manufacturing method called Selective Deposition Lamination (SDL) blends aspects of 3D printing and conventional lamination. SDL creates three-dimensional objects by selectively depositing and laminating layers of material. SDL usually uses sheets or rolls of material that are already solid, as opposed to some other 3D printing techniques that add material in a semi-liquid or powder condition [63]. Metals or polymers are two examples of the biomaterials that can be used to create these sheets. First, a digital model of the final product is created, and it is then divided into thin layers of cross-section. A cutting or stamping tool is used to cut each layer from the proper material, which is subsequently placed onto the build platform. The materials employed determine whether heat, pressure, or glue is utilized to adhere the subsequent layer to the preceding layer. Layer by layer, this process is carried out until the thing is finished. SDL has benefits for tissue engineering, including its ability to fabricate intricate structures with precise mechanical characteristics and high resolution. SDL-based bioprinting research is progressing quickly, with an emphasis on improving cell viability and functionality, optimizing process parameters, and investigating new biomaterial formulations for tissue regeneration applications [63].

### 2.4. Ultrasonic Additive Manufacturing

Tissue engineering can benefit greatly from the special capabilities of Ultrasonic Additive Manufacturing (UAM, Fabrisonic, OH, USA), which offers a flexible platform for creating intricate scaffolds and structures with specific qualities that promote tissue regeneration. Biocompatible materials related to tissue engineering, such as hydrogels, biodegradable polymers, or bioactive ceramics, can be joined using the ultrasonic bonding of thin metal layers that is the basis of UAM’s operation [64]. Utilizing its capacity to form complex structures without the need for melting or high temperatures, UAM makes it possible to precisely deposit a variety of materials, facilitating the incorporation of cells, growth factors, and biomolecules into the scaffold matrix. The ability of UAM to create scaffolds with regulated porosity, interconnectivity, and mechanical characteristics that replicate the natural extracellular matrix (ECM) architecture of tissues is one of its benefits in tissue engineering. Tissue regeneration processes are aided by these scaffolds, which offer a favorable environment for cell adhesion, proliferation, and differentiation. Additionally, UAM provides reproducibility and scalability, allowing for the creation of organ models and implants customized for each patient based on their unique anatomical needs. The main goals of current UAM for tissue engineering research include process parameter optimization, mechanical and biocompatibility characterization of UAM-produced scaffolds, and investigation of cutting-edge regenerative medicine applications.

According to recent research, UAM can be used to create scaffolds with bioactive properties and hierarchical structures that encourage tissue growth and cell adhesion [64]. The use of UAM in conjunction with biofabrication methods like electrospinning and bioprinting has also been studied in order to produce hybrid scaffolds with improved bioactivity and regenerative potential [64].

### 2.5. Digital Light Processing

One new bioprinting method that shows great potential for producing complex tissue structures and organ models quickly and with excellent resolution is digital light processing, or DLP. A digital micromirror device (DMD) is used in the DLP process to selectively pattern light onto a vat of photosensitive resin, which is then solidified layer by layer to produce three-dimensional structures. As the printing material, biocompatible hydrogels or bioinks containing cells and growth factors are frequently used in bioprinting applications [65,66]. In accordance with the computerized design, the DMD precisely regulates light projection, curing the photosensitive resin in designated locations. DLP has various benefits for bioprinting and tissue engineering. It makes it possible to quickly fabricate intricate scaffolds with sensitive features and great resolution, producing tissue structures that closely resemble the natural microenvironment. Additionally, DLP provides scalability by boosting efficiency and throughput by producing many structures at once within the same build volume. Additionally, DLP enables the scaffold matrix to contain a variety of materials and bioactive substances, which promotes cell adhesion, proliferation, and differentiation. Recent research has shown that with the created materials printed using the DLP process, it is feasible to create tissue scaffolds with complicated geometries that are appropriate for a variety of tissue regeneration applications and have strong biological features [65]. Furthermore, Greant et al. recently used the DLP approach to create intricate thermoresponsive structures for tissue engineering purposes. This demonstrates its suitability for bioprinting scaffolds for tissue engineering [66].

### 2.6. Direct Ink Writing

Another cutting-edge bioprinting method that provides fine control over bioink deposition to create intricate three-dimensional tissue structures with adaptable architectures is direct ink writing (DIW) (https://diw3d.com/, accessed on 12 February 2025). The idea behind DIW is to use computer-aided design (CAD) software (https://www.autodesk.com/blogs/autocad/autocad-2024/, accessed on 12 February 2025) to guide the extrusion of bioinks through a small nozzle or syringe tip onto a substrate, creating complex structures layer by layer. In bioprinting, bioinks are usually made of a viscous hydrogel matrix that contains biomolecules, growth factors, and cells. Controlled extrusion and shape retention during printing are made possible by carefully adjusting the bioink’s rheological characteristics [67].

For tissue engineering and bioprinting, DIW provides a number of benefits. The creation of heterogeneous tissue structures with spatial control over cell distribution is made possible by the ability to deposit different materials and cell types within the same construct. Additionally, DIW provides design diversity by producing scaffolds with pore sizes, geometries, and mechanical properties that may be tailored to meet the needs of various tissues and organs. Additionally, DIW promotes tissue regeneration, cell adhesion, and proliferation by making it easier for bioactive substances to be incorporated into the scaffold matrix. The DIW approach is now being used in several investigations. This method may use a separate system, which could be screw-based or piston-based. Systems powered by a screw provide better spatial control and the ability to release bioink at higher viscosities, while systems powered by a piston enable continuous control of the bioink flow [68]. In the liquid phase, using bioink lessens nozzle blockage. Typically, the smallest nozzle size is greater than 100 μm. This can result in an increase in shear stress, which eventually damages or kills cells [67,69]. A metallic tissue scaffold with macropores was created by Kachit et al. using a direct ink writing process [70]. Like this, Xu et al. used DIW to create a biodegradable Fe scaffold with different porosities. The creation of innovative tissue engineering materials employing DIW is thus the primary focus of numerous studies [71].

### 2.7. Liquid Deposition Modeling

A new bioprinting method called Liquid Deposition Modeling (LDM) has special potential for producing 3D tissue structures with great resolution and adaptability. Using a precision dispensing machine, liquid bioinks or hydrogels are deposited onto a substrate in the LDM process. There, they are solidified by solvent evaporation or cross-linking to create solid structures. Bioinks that contain cells, growth factors, and biomolecules are frequently employed as the printing material in bioprinting applications. By precisely controlling material deposition, LDM technology makes it possible to create intricate geometries and multi-material structures with specialized mechanical and biological qualities [72].

### 2.8. Extrusion-Based Bioprinting

A semi-solid substance, like a paste or gel, is extruded under pressure through a moveable nozzle or needle and deposited on a platform in extrusion-based printing, which is also referred to as semi-solid extrusion printing, pressure-assisted microsyringe (PAM) printing, or direct ink writing (Figure 1B) [73,74,75]. Similar to this, with FDM, the extrusion nozzle both slides up or the build platform slides down to deposit the subsequent layer when the first layer is finished. The final products’ performance is determined by the printing conditions and the materials to be printed, also known as inks [76].

In extrusion-based 3D printing, the viscosity and gel-forming properties of materials are crucial [77]. Shear-thinning inks and quick recovery times are generally needed [78]. In other words, materials should be just viscous enough to hold their shape while printing, but not too viscous as to prevent nozzle blockage and produce uniform strands. Rapid gelation of the materials is necessary after printing to prevent the printed construct from being altered [79,80]. Chemical cross-linking, crystallization, chain rearrangement, and the creation of hydrogen bonds and non-covalent interactions can all lead to solidification. The procedure of extrusion offers compatibility with an extensive variety of materials, including hydrogels, which have drawn increased attention due to their precisely regulated viscosity, molten polymers, pastes, and polymer solutions [81].

The main difficulties in extrusion-based printing are preserving the printed construct’s structural integrity and attaining a high level of printing fidelity. Viscosity is not the only crucial characteristic that needs to be tuned; the material output, needle diameter, nozzle feed rate, printer head speed, viscosity of extruded ink, temperature, and length between the nozzle and substrate are also critical factors. Each of these factors is highly dependent on the others. For example, reducing the needle diameter will result in a smaller strand, and higher pressure may be required to keep the strand deposition consistent [82].

The capacity to co-print inks made of different polymers, which makes it simple to access spatially controlled multi-material scaffolds, and the ability to print a variety of inks with a wide range of viscosities are only two advantages of extrusion-based 3D printing. Furthermore, this printing method is particularly suitable for printing scaffolds filled with drugs or cells because it uses lower temperatures than FDM.

### 2.9. Stereolithography (SLA)

SLA is based only on the photopolymerization hypothesis. Free radicals are created when the photoinitiator and UV light interact [83]. When the components of the photosensitive liquid resin are subjected to a UV laser, they undergo targeted polymerization [84]. UV light aimed perpendicular to the liquified resin’s surface can pass through it in SLA. After the deposit has set, a second coat of liquid resin is applied. Until a final product is created, this procedure is repeated. When a particular product is completed, excess resin is pumped away for future use. The final product is cleaned to get rid of extra resin. After that, the support assemblies are divided. Like the cast pieces, the printed item usually has rough edges that are further polished with a coating [85,86]. Compared to other manufacturing methods, laser-based manufacturing is more demanding because it produces high-resolution parts, improves surface quality without the need for post-processing, increases z-axis growth because printed layers bond the best, and requires less time [87]. The most important factor in SLA printing is the thickness of the cured layer, which is affected by light energy [88].

### 2.10. Selective Laser Sintering (SLS) and Selective Laser Melting (SLM)

The powder category of additive manufacturing includes SLM and SLS, which require a high-intensity laser for the ensuing coalescence of powder particles. The only distinction between SLS and SLM is the intensity of the scanning. SLS uses limited melting, whereas SLM uses high-intensity entire melting, which densifies the material. The flattened bed receives the ideal laser energy dose. At the same time, the powder particles are processed using SLS thanks to the carefully controlled movement of mirrors and lenses. The temperature is quickly raised to the necessary depth by a focused laser beam, sufficient to cause necking between nearby powder particles and ensure the sintering phenomenon. Densification is directed by sintering, which can alternatively be started by binder melting to bind powder particles using heat energy from a laser [89].

By the use of elevator motion, the sintered layer is descended. A fresh powder layer is rolled to prepare for the subsequent sintering cycle. The cycle repeats itself until the predefined design item is printed. To prevent the unfavorable effects of the surrounding gases, the whole printing process in the SLS chamber is carried out in an inert environment [90]. It is not necessary to utilize auxiliary supports because the unsintered powder can be used as a support to remove complex designs later during the post-processing step after printing. An efficient coalition of powder particles is required to construct the models intended for various self-propelled vehicle applications, including transportation, aircraft, space applications, spacers for water filtration, and a revolutionary approach in medicine [91,92,93,94].

For biomedical implantations using stainless steel 316L in physiological environment functioning, improved mechanical properties in Ti_6_Al_4_V alloys, including tensile strength, ductility, fatigue resistance, and other surface qualities, were also documented [95,96,97]. In addition to this advantage, the short-lived confined heat raises concerns about the printed object’s precision. It provides lingering stresses in the freshly produced powder bed that result in deformities and render the printed product ineffective [98]. On the other hand, 4D printing purposely employs regulated residual stress, which acts as a driving force to generate programmed conformation. The production of thin stainless steel 316L and the use of AlSi_10_Mg both validated the unfavorable residual stress in 3D printing [99,100].

### 2.11. Inkjet or Binder Jet Printing

An additive manufacturing method called binder jetting has great potential for creating organ-like structures and tissue scaffolds in bioprinting. A liquid binding agent is selectively deposited onto a powder bed of biocompatible materials, like hydrogels or ceramics, as part of the binder jetting process [101]. This procedure is managed using a computer-aided design (CAD) model that directs the layer-by-layer deposition of the binding agent in accordance with the object’s intended geometry. Following the deposition of one layer, another layer of powder is applied to the construction platform, and so on, until the entire thing is created. Following printing, the construct is subjected to post-processing procedures like curing or drying in order to eliminate surplus powder and harden the binder. For tissue engineering applications, binder jetting has a number of benefits. By simulating the natural ECM architecture of tissues, it enables the creation of intricate, porous scaffolds with pore sizes, shapes, and distributions that can be customized [102].

Furthermore, binder jetting makes it possible to include cells, growth factors, and bioactive chemicals into the scaffold matrix, which encourages cell adhesion, proliferation, and differentiation. Additionally, binder jetting is a flexible and scalable manufacturing process that can create organ models and implants customized for each patient based on their unique anatomical needs [63]. Optimizing printing parameters, describing the mechanical and biological characteristics of printed constructions, and investigating cutting-edge tissue regeneration applications are the main goals of current research in binder jetting for bioprinting.

Zhou et al. have recently shown that binder jetting is feasible for creating scaffolds with superb geometric accuracy using hydroxyapatite powder (HA) and water-soluble glue [102].

### 2.12. PolyJet Printing

PolyJet is a sophisticated bioprinting method that produces precise and high-resolution three-dimensional tissue structures by layering photopolymer droplets using inkjet technology. According to the PolyJet concept, liquid photopolymer materials are jetted onto a build platform, where they are immediately cured by UV radiation to create solid structures. Bioinks that contain cells, growth factors, and biomolecules are frequently employed as the printing material in bioprinting applications [63]. Multiple materials with different properties can be deposited simultaneously thanks to PolyJet technology, resulting in heterogeneous tissue structures with spatial control over material composition. It has various benefits for bioprinting and tissue engineering. The creation of intricate scaffolds with anatomical precision and microscale structures that closely resemble the original tissue microenvironment is made possible by its high resolution and precise feature detail. PolyJet facilitates the integration of various cell types and bioactive substances into the scaffold matrix, thus fostering tissue organization, cell–cell interactions, and functional integration. Additionally, PolyJet allows for the customization of mechanical qualities, degradation kinetics, and bioactivity, as it can print with a variety of biocompatible photopolymers [103]. Table 2 provide comparison of each different methods with advantages and disad-vantages.

## 3. Bioink

In 2003, the phrases “bioink” and “bio paper” were first used in relation to organ printing [104]. One of the key components of 3D bioprinting is bioink, a mixture of bioactive substances, biomaterials, and cells. It creates interesting printed scaffolds. In order to use living cells or tissue spheroids as the “bioink” for bioprinting, a bio paper (hydrogel) was first intended to be supplied or printed. As a result, the term “bioink” originally referred to a biological material that was applied to or integrated into three-dimensional hydrogels. Cells and cell collections were utilized as the bioink in a large amount of the initial 3D bioprinting research [105,106]. Some experts claim that even at this early stage, a feasible bioink composition needs to be “functionally and structurally” further refined. It preserves shape fidelity, supports cellular processes, and matches tissue mechanical qualities to provide stability and biocompatibility. For biomedical applications such as tissue engineering and regenerative medicine, this optimization is essential to producing bioprinted tissues that are viable, functional, and reproducible. During this time, numerous studies have been conducted on various types of bioink, and the range of additive manufacturing techniques available for bioprinting has increased [104].

Supporting, escaping, structuring, and operating bioinks are the four categories into which the term “bioink” has recently been separated [107]. After delivery, supporting bioinks act as an artificial extracellular matrix (ECM) as the cells divide and maintain cell populations. Internal channels or voids can be quickly created by quickly removing escaping bioinks from a 3D bioprinted structure. Structuring bioinks are used to provide mechanical stability to the printed scaffolds [104]. During 3D bioprinting, functional bioinks provide the scaffold with biochemical, mechanical, and electrical signals that affect biological behavior. For a short while, they might be fleeing (like polycaprolactone thermoplastics). This classification of bioink is based on the component materials rather than the fabrication process used to generate it because of the importance of essential components in the final functionality of a 3D bioprinted scaffold [104].

Rheological properties are the physicochemical elements that most significantly affect hydrogels’ printability [108]. The study of rheology examines how materials alter their motion and structure in response to forces [79]. A bioink that is initially in a bulk resting state undergoes a transformation as it passes through the nozzle, changing to a greater shear form before ultimately reaching a new idle state in 3D extrusion-based processes. The main rheological characteristics of these transitions include viscoelastic shear moduli, viscosity, shear stress, and elastic recovery [109].

### 3.1. Viscosity

Viscosity, or a fluid’s resistance to flowing under stress, has a big influence on the print accuracy and efficacy of cell encapsulation. Higher printing quality is usually the result of improved viscosities. However, high viscosity also results in increased shear stress, which may affect the cells that float in the bioink. The molecular weight and concentration of a polymer in solution are the two main variables influencing its viscosity [79]. The shear stress to shear rate ratio is used to compute viscosity. Newtonian fluids are considered to have a linear relationship between shear stress and shear rate. In non-Newtonian fluids, ratios deviate from linearity and might either be growing or decreasing. Shear thinning and shear thickening are examples of time-dependent and time-independent fluids that fall under the category of non-Newtonian fluids (including rheopectic and thixotropic solutions) [110].

### 3.2. Shear Thinning

Shear thinning is the most common time-independent non-Newtonian fluid phenomenon, which happens when shear rates are increased and viscosity is decreased. This property is commonly seen in materials used in extrusion printing, such as polymer melts, hydrogels with partial cross-linking, gels, and solutions of polymers [110].

During the extrusion printing process, when shear pressures greatly increase, shear thinning refers to a bioink’s ease of extrusion and preservation of its original shape while having a lower viscosity [7]. Following extrusion, the viscosity increases, and the shear rate drops, helping to maintain the printed shape. While waiting for secondary cross-linking to occur, the printed structure will not break down because of the continuous flow and mild distortion caused by a higher zero-shear viscosity. For example, the high zero-shear viscosity of printing calcium phosphate cement helps to maintain shape [111].

The physicochemical interactions that underlie form preservation and shear thinning in different types of bioinks are caused by different molecular mechanisms. At repose, higher weight polymer molecules are randomly oriented and entangled. The polymer bonding untangles and aligns as a result of the shearing, reducing viscosity and internal resistance. These bioinks’ abrupt transition from a liquid to a solid state results in form retention [112].

When shear-induced breaking of solid particle contacts takes place, bioinks based on colloidal solid suspensions, dispersions, and pastes experience shear thinning. Shape stability is made possible by the comparatively high viscoelasticity at rest that results from the restoration of the connections between the suspended particles [113]. Notable examples of this category include the use of calcium phosphate cement and polymer solutions, such as nano-silicate dispersions in biomaterial inks or bioinks. These bioinks’ setting reaction also helps with shape fixing after printing [113]. Another class of non-time-dependent fluids are shear thickening materials. In non-Newtonian materials, viscosities could not be shear-dependent but rather time-dependent. The viscosity of thixotropic materials steadily drops at a steady shear rate until, following a period of rest, it stabilizes at its initial value. In contrast to viscosity, which rises with time, shear rate characterizes rheopectic behavior [110].

### 3.3. Yield Stress and Viscoelastisity

Using biocompatible hydrogels or bioinks, extrusion-based 3D bioprinting has become a viable technique for creating functional tissue structures. Because of a special set of characteristics, these bioinks may pass through tiny nozzles with little resistance and maintain their shape after deposition. Because the shear pressures produced during extrusion can negatively impact the viability and functionality of living cells, this characteristic is especially crucial when printing with them. Bioinks must have viscoelastic behavior—that is, the capacity to demonstrate both viscous flow (i.e., the ability to flow) and elastic deformation (i.e., shape retention)—in order to attain excellent flow and shape-retention capabilities. The storage modulus (G′) and the loss modulus (G″) are two important metrics that can be used to describe the viscoelastic characteristics of bioinks. The amount of energy that is elastically stored during deformation is measured by the storage modulus, which is closely related to the material’s capacity to maintain its shape. The loss modulus, on the other hand, indicates the material’s flow characteristics and quantifies the quantity of energy lost as heat during deformation.

The viscoelastic properties of bioinks must be carefully controlled when designing them for extrusion-based 3D bioprinting in order to guarantee that they can flow easily through the printing nozzle and maintain their shape and structure after deposition. To accomplish this, a comprehensive understanding of the interactions between the printing parameters—such as extrusion pressure, printing speed, and nozzle size—and the rheological properties of the material, such as its viscosity, shear modulus, and elasticity, is necessary. A crucial metric that measures the quantity of energy lost as heat during deformation of a material is the loss modulus G″. The material’s flow characteristics, which are essentially distinct from its resistance to deformation, are closely related to this parameter [114].

Oscillatory rheology, which entails exposing the material to oscillating forces and observing its response, can be used to study viscoelasticity, whereas viscosity is a measure of a material’s resistance to flow under rotational forces. It is standard procedure to compute the storage modulus G′ and loss modulus G″ as functions of oscillation frequency and amplitude in order to completely describe the viscoelastic behavior of a bioink. One important measure of the material’s flow is the loss tangent, sometimes called the damping factor (tan(δ)), which is the ratio of G″ to G′. The yield point, in addition to the storage and loss moduli, is a crucial parameter that can be used to characterize the material’s capacity to maintain its shape [115]. The smallest amount of stress required for a material to undergo deformation is known as the yield point. The material’s total mechanical strength and form retention characteristics are determined by its cross-linking density and elastic modulus, which are closely related to this yield stress. By providing internal resistance to shape change, these interactions enable the material to perform as an elastic solid, returning to its original shape in the event of little perturbations [115]. However, the material undergoes irreversible deformation at a certain threshold, such as yielding or flow points. The yield point can be estimated using rotational rheology by using the viscous properties. The measurement’s susceptibility to instrument bias is this method’s drawback. Therefore, oscillatory rheology is recommended [111].

The higher yield stress of an ink frequently improves filament manufacturing and design stiffness, but it can also make cell encapsulation more challenging. Typically, additives like gellan gum, hyaluronan, or carrageenan increase the yield stress of a particular ink [116,117]. In the presence of a positive ion in solution, gellan gum is added to gelatin methacryloyl to form a shear-reversible, ionically cross-linked network, which increases the viscosity of the ink at rest. Shear forces damage this reversible network during dispensing, and when the shear forces stop, the network heals itself [117]. A deposition of material with the required yield stress or elastic characteristics will not occur if the acting forces are less than this yield threshold value.

These forces include surface tension, capillary forces, and gravity, which is based on the weight of the filament and all levels above it [109]. Another crucial idea is the transition from fluid-like movement to elastic form preservation. This characteristic can be quantified by tracking the recovery of the viscosity or shear moduli over time after loads greater than the yield are eliminated [118,119,120]. Although lithography-based bioprinting and extrusion printing have rather different rheological requirements, they are both essential for determining printability and shape accuracy in those processes. These rheological characteristics include shear-thinning behavior and a quick, reversible sol-gel transition. When each layer is cross-linked in dynamic light processing (DLP) and stereolithography (SLA), a new volume of the materials must flow smoothly beneath the construction platform, ideally even without a mixing device [121].

## 4. Polysaccharides

### 4.1. Cellulose

As the most common natural organic polymer in the world, cellulose is renowned for its environmental friendliness, biodegradability, renewability, and biocompatibility [122,123]. Because of its many reactive groups, cellulose is a biomaterial that is easy to cross-link and change. It is made up of linear chains of β (1→4)-linked D-glucose units [124]. However, each structural unit of cellulose has three hydroxyl groups, and the large number of hydroxyl groups in cellulose makes it extremely hydrophilic. Furthermore, cellulose’s poor solubility in common solvents is caused by the hydroxyl groups’ propensity to form hydrogen bonds with water. By substituting an oleoyl group for the hydroxyl group in the molecular chain, Yang et al. chemically altered cellulose using an acoustic-chemical approach, improving the material’s solubility and processability [125].

Cellulose can fulfill a variety of purposes and requirements after being processed and altered. Cellulose ether and cellulose ester-based cellulose ether ester are the three types of cellulose derivatives that can be separated based on the structure of the changed product. These are the more prevalent cellulose derivatives: carboxymethyl cellulose (CMC), hydroxypropyl methylcellulose (HPMC), hydroxyethyl cellulose (HEC), and hydroxypropyl cellulose (Figure 2). HPMC’s nontoxicity and swelling characteristics make it a popular choice for controlled drug release applications.

To aid in wound healing, a thermoresponsive in situ hydrogel based on HPMC releases medications when heated on the skin’s surface [127]. Designed as cellulose-based hydrogels, HEC and CMC are widely employed in antibacterial, wound healing, and drug delivery applications due to their high levels of biocompatibility, water solubility, and biodegradability. Furthermore, numerous researchers have created cellulose derivatives with antibacterial activity by cationizing cellulose nanostructures, while cellulose itself lacks antimicrobial qualities. Using N,N,N0,N0-tetramethylethylenediamine (TEMED) as a catalyst, Zhu et al. created cellulose nanofiber (CNF)-reinforced poly(Ionic Liquids) (PILs) composite hydrogels (PACxVy) in a single step. The hydrogels that were produced showed notable temperature sensitivity. These substances might be employed as sensors of wound temperature. The addition of PILs with antibacterial qualities gave the hydrogels strain sensitivity and bacteriostatic capability [128].

### 4.2. Chitosan

Deacetylated chitin is typically the source of chitosan, a necessary natural polymer with macromolecular chains abundant in amino, hydroxyl, and other functional groups. After cellulose, it is the second-largest natural renewable polymer [129,130]. Chitosan is widely used in biomedicine and other domains because to its excellent antibacterial qualities, low toxicity, and ease of molecular modification. Chitosan is frequently utilized in drug delivery because it can be loaded into hydrogels based on chitosan with medications of different molecular weights that can be broken down and metabolized in vivo following drug release [131,132]. In practical usage, chitosan-based stimulus-responsive hydrogels have a number of drawbacks, including low chitosan water solubility, poor water resistance, and poor mechanical qualities [133,134,135]. The primary cause of chitosan’s low water solubility is because hydrogen forms a stiff crystal structure both inside and between molecules, and the amino groups on the molecular chain are not ionized in their normal state. The production and study of chitosan derivatives has attracted the attention of numerous researchers. Chitosan is altered chemically to improve its functional qualities and increase its range of uses [136]. To increase chitosan’s crystallinity and water solubility, alkyl halides are utilized to substitute the amino or hydroxyl groups on its molecular chain. Additionally, chitosan is altered by a Schiff base reaction to produce amphiphilic chitosan [137,138,139]. Chitosan oligosaccharide, which is produced by breaking down chitosan, can be directly soluble in water, control blood sugar and blood sugar levels, and regulate the body’s immune system. Carboxymethyl chitosan (CMCS), which is produced by chemically altering chitosan, has improved water solubility and maintains the antibacterial qualities of chitosan [140,141,142,143].

The QCMCS + OHA + DTP injectable self-healing hydrogel (QOD Hydrogel) was created by Zhang and colleagues using quaternized carboxymethyl chitosan (QCMCS), oxidized hyaluronic acid (OHA), and 3,30-dithiobis-(propionohydrazide) (DTP) [144]. While QCMCS, OHA, and DTP interacted to form imine, hydrazide, disulfide, and hydrogen bonds, the application of CMCS gave the hydrogel some antibacterial qualities. The network of hydrazide and disulfide linkages inhibits the hydrogel’s quick hydrolysis in acidic settings, while the imine and hydrazide bonds’ inertness stabilizes the hydrogel in non-acidic situations. The imine, arylhydrazone, and disulfide linkages’ dynamic reversibility gives the hydrogels their capacity for self-healing as well as a variety of pH-responsive characteristics. To verify the hydrogels’ drug-loading and release properties, the authors did not, however, carry out any in vivo tests. Tannic acid (TA) and CMCS were used by Li and colleagues to create a multifunctional bilayer hydrogel [145]. Formyl phenyl boronic acid’s bifunctional group connected TA and CMCS to create biodegradable hydrogels in the inner layer. The hydrogels’ pH-responsive behaviors were made possible by the presence of dynamic covalent bonding, which facilitated the accurate and real-time detection of wound pH levels as well as the encouragement of wound healing.

### 4.3. Alginate

Large brown algae are the primary source of alginic acid, which is used to make alginate. It has low manufacturing costs and good gel qualities. It is typically found as cationic salts, including magnesium (Mg^2+^), barium (Ba^2+^), sodium (Na^+^), and others [146,147,148]. Sodium alginate is the most widely used alginate agent. A linear natural polymer comprising α-1,4-L-guluronic acid (G) (α-L-guluronic, G) and β-1,4-D-mannuronic acid (M) (β-D-Mannuronic, m) units make up sodium alginate. Its molecular chain contains several hydroxyl and carboxyl groups [149]. When sodium alginate reacts with different divalent or trivalent metal cations in an aqueous solution, sodium alginate hydrogels can be created [150]. Alginates’ sensitivity to pH makes them popular for medication administration. Due to alginates’ insolubility, the hydrogels’ swelling capacity and drug release are both reduced in non-acidic environments. In contrast, the hydrogel performs better in a neutral buffer medium in terms of swelling and drug release.

Benzyltrialkylammonium (BTA) particles loaded with pyran were enclosed by Lynne’s group in calcium cross-linked alginate hydrogels. The BTA particles in the hydrogels showed a robust fluorescence response to raised pH in the pH range of 6–9. Calcium cross-linked alginate hydrogels have been authorized by the U.S. Food and Drug Administration to be used as wound dressings for diabetic foot ulcers (DFUs). Alginate hydrogels’ nanoporous shape efficiently holds onto BTA particles, and alginate is a very biocompatible and immunogenic material. By using a fluorescent reaction to measure the pH of the wound, this hydrogel system is anticipated to enhance DFU diagnosis, staging, and therapy response evaluation [151]. On the basis of alginate and poly (propylene glycol) 407 (F127), a Ce6-loaded MOF thermosensitive hydrogel (Ce6@MOF-Gel) was created. This hydrogel offers a novel approach to hastening the healing of infected wounds because of its pH-responsive release and photodynamic response [152]. Adenosine triphosphate (ATP) is released in large quantities during the metabolism of tumor cells. When Sun et al. combined alginate with an ATP-specific aptamer (Aapt), an alginate-based hydrogel was created because the tumor contained endogenous calcium ions. The immunoadjuvant CpG oligodeoxynucleotides were released from the hydrogel in response to the release of ATP from tumor cells, which enhanced the effects of radiation or chemotherapy in a complementary manner [153].

### 4.4. Pectin

The usage of pectin in the pharmaceutical and biotechnology sectors is growing. This naturally occurring polysaccharide is essential to the structural integrity of plant cell walls. Unlike other polysaccharides, pectin is safe to consume and decomposes spontaneously. However, there are a number of drawbacks to pectin formulations, including inadequate drug loading, limited mechanical strength, poor shear stability, and early drug release [154]. Pectin is a poly α-1-4-galacturonic acid with varying degrees of carboxylic acid residue methylation. Pectin, which possesses strong gelling properties, is found in several citrus byproducts. Pharmaceutical businesses have utilized pectin alone or in conjunction with its gelling properties to treat and improve health [155]. Ranjha et al. used pectin and acrylic acid to create pH-sensitive hydrogels in order to investigate the release of verapamil. Using benzoyl peroxide and N, N′-methylene bisacrylamide (MBA) as initiators, pectin and acrylic acid were combined to create the hydrogels [156].

### 4.5. Hyaluronic Acid

HA is a naturally occurring polysaccharide with a variety of sources and strong biocompatibility [157]. One of the essential parts of the human body, HA is derived from animal tissues like chicken crowns and bovine vitreous bodies, in contrast to other polysaccharides [158].

Microbial fermentation, which entails direct extraction from animal tissues, has supplanted the preparation method of HA due to the quick development of biotechnology. The broad use of HA is determined by the simplification of preparation technology [159,160]. Over time, HA has emerged as a popular biomaterial for making hydrogels, and the range of medical goods made from it has grown. To create a polysaccharide gel dressing that may mimic the skin structure, Xiong’s research team utilized oxidized HA and CMCS. When exposed to NIR radiation, the hydrogel exhibits a good warming effect. Hydrogel-loaded gentamicin’s warming effect and synergistic sterilization offer a practical way to prevent and treat infected wounds [161].

A crucial part of the joints and eyes, HA has been extensively researched in tissue engineering and drug delivery [162,163,164]. To create stimulus-responsive hydrogels with improved therapeutic efficacy and less harmful side effects, numerous investigations are dedicated to mixing HA-based hydrogels with a range of medications [165,166]. In order to treat tumors effectively and precisely, Dai and colleagues created an ultrasound-mediated hydrogel delivery platform (HFTiDP) that can react to exogenous ultrasound stimulation to alter the expression of drug-resistant genes or proteins [167]. Based on thiol HA (SH-HA) and disulfide hyperbranched polyethylene glycol (HB-PBHE), a multifunctional hydrogel (HA@Cur@Ag) was created, and curcumin lipids were loaded into the hydrogel together with silver nanoparticles (AgNP). This hydrogel has dual antibacterial, anti-inflammatory, and antioxidant qualities that can help diabetic wounds heal more quickly and reduce inflammation and oxidative stress [168]. The technology for HA preparation and application is advanced. Its applicability is still constrained by a few issues, though. Because of its extreme hydrophilia and instability, some experts think that HA may worsen inflammation or encourage the spread of cancer cells [169,170]. The effect of the critical molecular weight of HA on tumors is yet unknown, despite the fact that physical and chemical changes can improve the physical characteristics of HA and that regulating its molecular weight can influence tumor cells. As a result, it is critical to steer clear of disputes and work toward taking a special part in promoting hydrogel reactions [171].

### 4.6. Starch

Starch, a neutral polysaccharide, is produced by plants such as rice, wheat, and maize as insoluble granules that are their main energy source [172]. Amylopectin, a branching α-d-glucan with a molecular weight of 106–107 g mol^−1^, and amylose, a linear chain composed of (1,4)-linked α-d-glucan, are the two different polysaccharides that make up starch, excluding its plant source. This polysaccharide creates a suspension when it dissolves in water at room temperature. The grains, however, swell and turn gelatinous when heated. This aids in the formation of a continuing condition around the more noticeable granules and the separation of the amylose part from the amylopectin.

The amylose phase separates and gels when the starch solution cools. The gel can be used in drug delivery systems in the biomedical industry because of its exceptional stability and biocompatibility [173]. We are aware of no previous use of starch to produce cell-loaded bioinks for 3D bioprinting applications. One explanation could be the discovery by Aljohani et al. that heat treatments are required to maintain the integrity and dehydration of structures printed using starch [174]. The potential of modifying cassava starch by an ozone process has been examined by Maniglia and associates in order to ascertain whether the starch may be used to produce hydrogels for 3D food printing [175]. They developed a modified starch that is appropriate for extrusion 3D printing based on the degree of ozonation. They were, nevertheless, smaller molecules with higher carboxyl and carbonyl contents and an acidic nature. This altered starch produced hydrogels with different properties. A separate technique was used by Noè et al. to produce a hydrogel that could be photo-cross-linked using starch methacrylate [176].

Its processability was evaluated using digital light processing (DLP) 3D printing or photocuring in a mold. By using this modification technique, scientists could produce hydrogels with the right mechanical and rheological properties and print starch structures without the need for additional heat treatment. Similarly, 3D-printed scaffolds for Schwann cell seeding had noticeable printing gaps when gellan gum and pure starch were combined [177]. The results indicated that the printed structures did not harm the L929 fibroblast cell line, were stable, and displayed the proper swelling ratios. Consequently, starch will be more efficiently utilized to produce cell-rich bioinks for later applications. There, too, the same methods may be used. Several polysaccharides have been the subject of 3D bioprinting research.

### 4.7. Glucan and Its Derivatives

Algae, barley, yeast, and mushrooms all contain the fiber carbohydrate glucan [178,179]. Pectin’s structural chains are rich in hydroxyl groups, which facilitates cross-linking and modification of the protein through chemical or physical means. Glucan controls blood sugar, lowers cholesterol, and boosts immunity, among other things. It is also a perfect medicinal material because of its high hydrophilicity and chemical inertness [180,181]. Using CMCS and oxidized dextran, He and colleagues created multifunctional hydrogels (COO@PRP hydrogels) loaded with platelet-rich plasma (PRP) [182]. The COO@PRP hydrogel has dynamic Schiff base and hydrogen bonds, which enable the hydrogel to respond to pH and ROS in two ways. This allows for the controlled release of growth factors to aid in the healing of skin wounds. Apart from conventional exogenous reactions, a hydrogel based on dextran that reacts to modifications in the bacterial infection’s microenvironment was created [183]. Chitosan modified with L-arginine and oxidized dextran modified with phenylboronic acid form the basis of the hydrogel system, which reduces oxidative stress, regulates the healing process, encourages the remodeling of the wound microenvironment, and speeds up skin regeneration.

### 4.8. Agarose and Its Derivatives

Red algae produce agarose, a polymer that has the ability to gel in a thermally reversible way. At temperatures above 90 °C, it dissolves in water, while at temperatures below 40 °C, it takes on the consistency of gel. Because of its hysteretic qualities, stability, and ease of water absorption, agarose can be used as a gelling agent in a variety of industries, including food, textiles, and biomedicine. Using agarose and polyacrylamide, Xiang and colleagues created a double network (DN) structural hydrogel that responds to NIR light [184]. In order to give the hydrogel NIR responsiveness, photothermal agents were added. This work produced new concepts for biomedical applications by achieving quick gelation and efficient, controlled drug release.

### 4.9. Dextran

Bacteria employ dextran, an exopolysaccharide produced by a variety of bacteria, to form biofilms, which are protective microbial coatings. Leuconostoc meenteroides, Lactobacillus brevis, and Streptococcus mutans are a few of these bacteria. The molecular weight of the branching polysaccharide dextran ranges from 1 to 40,000 kDa. α-(1,4) or α-(1,2) branching linkages join its glucose units intermittently, followed by α-(1,6) and α-(1,3) connections in order. Hydrogels cannot be made from pure dextran; it needs to be chemically altered by oxidation or functionalization with methacrylate groups to enable cross-linking [185].

One study bioprinted a vascularized construct for wound therapy using oxidized dextran and a possible core/shell extrusion-based 3D bioprinting technique. A hydrogel comprising peptide-functionalized succinylated chitosan (C), periodate-oxidized dextran (D), and GelMA as the core was employed. Two different cell types were used in this experiment: BMSC in the shell and HUVECs in the core. The peptide-CD/GelMA structures that were 3D bioprinted significantly promoted cell differentiation and proliferation.

Twenty-one days after the bioprinting process, the tube-like constructs revealed markers specific to HUVEC endothelial cells. Osteogenic differentiation demonstrates that BMSC multipotency, which is required to develop regenerative constructs that can divide into several cell types when required, was unaffected by any of the designs. Although this work is the only one on the topic, it overcomes a major barrier to 3D bioprinting living tissues by paving the way for the creation of vascularized structures using bioinks based on dextran [186].

### 4.10. Xanthan Gum

The bacteria Xanthomonas campestris produces xanthan gum, a negatively charged exopolysaccharide with an average molecular weight of about 2000 kDa. Its primary chain is 1,4-linked β-D-glucose, and its trisaccharide side chains are β-D-mannose, β-D-glucuronic acid, and α-D-mannose [187].

The following are the xanthan gum’s temperature-dependent conformations in aqueous solutions: through gel-like activity, it displays a firm and ordered double helical strand structure at low temperatures (below 40–50 C); at higher temperatures (above 50 C), it displays a flexible and disordered coil shape. The fact that xanthan gum clumps when it dissolves in water due to inadequate hydration is one of its drawbacks [188]. This results in the formation of a gelatinous outer layer that keeps water out and stops the polysaccharide from fully degrading. To alleviate this issue, xanthan gum can be modified or mixed with other polymers to increase its water solubility [189]. Despite this, xanthan gum is a popular viscosity regulator due to its exceptional rheological qualities. This makes it an excellent method for improving the rheological and mechanical characteristics of bioink. However, its application in 3D bioprinting bioinks is still relatively new.

Lim et al. primarily used xanthan gum’s shear-thinning qualities to produce a suitable bioink for extrusion-based bioprinting. They combined xanthan gum, alginate, GelMA, hMSCs, and CMC in various concentration ratios [190]. Since the viscosity of the bioinks decreases as the shear rate increases, xanthan gum seems to have a role in the shear-thinning tendency. At the site of analysis, all formulations displayed gel-like behavior because the G′ was more significant than the loss modulus (G”). They evaluated UV + ionic cross-linking and UV irradiation as two different cross-linking methods. There were no differences in the viability of the hMSCS cells on the bioprinted structures cross-linked utilizing the two distinct techniques. However, when the cross-linked hydrogel structures were subjected to ionic and UV conditions, the bromodeoxyuridine test showed significantly more cell growth. Different pore sizes and distributions may affect the behavior and proliferation of cell infiltration, according to the scientists who provided an explanation for these data. Muthusamy et al. examined the material’s thickening properties by combining xanthan gum with neutralized collagen type 1 to generate a bioink [191]. To promote the formation of capillaries, this bioink may be used to bioprint endothelial cells in exact spatial arrangements across fibroblast layers. A higher viscosity and higher G’ in the rheological investigation showed that the collagen-based formulations were more printable after xanthan gum was added. Shear thinning and gel-like activity in the bioink were indicated by G′ values, which were more significant than G”.

### 4.11. Gellan Gum

The 500 kDa anionic extracellular bacterial polysaccharide gellan gum is produced by the microbial fermentation of Sphingomonas paucimobilis. A repeating unit of β-(1,3)-D-glucose, β-(1,4)-D-glucuronic acid, β-(1,4)-D-glucose, and α-(1,4)-L-rhamnose comprise it [192]. This polysaccharide is also thermoresponsive since it can remain in coil shape in solutions at temperatures above 60 degrees Celsius. Below this temperature, it forms a hydrogel and changes into a double-helix structure.

Additionally, because hydrogels have carboxylic groups in their structure, they can form in the presence of mono and divalent ions, such as Na^+^, K^+^, Mg^2+^, and Ca^2+^, respectively [193]. Since gellan gum hydrogels made via ionotropic cross-linking are brittle and mechanically weak, they usually require chemical modification or combination with other polymers to form printable bioinks. Using extrusion-based 3D bioprinting of MSC-loaded polylactic acid microcarriers encapsulated in gelatin methacrylate-gellan gum (GelMA-GG) hydrogel bioinks, the shear-thinning characteristics of gellan gum were first investigated in 2014 [194] for the biofabrication of live tissue constructions. Gellan gum was employed to boost the solution viscosity and enhance the bioink’s printability since GelMA is insufficiently viscous.

Gellan gum was added to the printed filaments to maintain their shape following deposition, enabling the creation of structures with excellent shape fidelity. Additionally, three days after bioprinting, MSC cell survival exceeded 90%, and the cells were uniformly dispersed throughout the hydrogel matrix. In a more recent study, Wu et al. [195] paired the fast photo-cross-linking of poly(ethylene glycol) diacrylate (PEGDA) with the shear-thinning and recovery characteristics of gellan gum to create a double network hydrogel that produced human-scale, very accurate human ear and nose structures. The behavior was liquid-like (G″ > G′) in large strains and gel-like (G′ > G″) in small strains for all formulations, which all exhibited shear-thinning properties. Molecules MC3T3-E1 and mouse BMSCs were added to these hydrogels. Both cell types maintained strength and stability throughout the 21-day cell culture period. Additionally, as seen by integrated optical density (IOD), the bioprinted scaffold created an open network with enough oxygen and nutrient exchange, encouraging cell activity. Similarly, to improve the printability of the inks and form integrity of the related structures, Zhuang et al. mixed GelMA with gellan gum as a viscosity booster [196].

Zhuang et al. created a novel printing technique that enables the creation of more complicated forms by utilizing UV-assisted extrusion printing technology. The printability investigation helped select six GelMA-GG ink combinations that showed effective cell encapsulation, little cell sedimentation, and acceptable printability. The gain in compressive modulus for this set of formulations from 9 to 16 kPa indicates that, as predicted, increasing polymer content increased the compressive modulus. The fact that there were more cells after 7 days indicates that the C2C12 cells proliferated more quickly when bioprinted in 5–0.5% (*w*/*v*) GelMA-GG instead of 7.5–0.5% (*w*/*v*) GelMA-GG. This might be because of the materials’ rigidity and more favorable micro-structure. Gellan gum derivatives containing methylacrylate can also be made.

### 4.12. Konjac Gum

Konjac glucomannan (KJG) is a water-soluble hydrophilic polymer that is extracted from konjac tubers. KJG has long been utilized as a traditional food and medicine in China, Japan, and Southeast Asia. The primary structural chain of KJG is polymerized by the α-(1→4)-pyranoside bonds between D-glucose and D-mannose, with less acetyl groups at C-6 positions in the side chain unit [197]. KJG is an attractive biopolymer due to its noteworthy characteristics, which include gelling ability, film formation ability, biodegradability, and biocompatibility [198]. KJG–xanthan gum hydrogel was introduced by Alves et al. for use as a wound dressing. Following that, physicochemical and biological tests were used to evaluate the dressing [197]. After its synthesis, Yang et al. examined the application of a KJG hydrogel in the management of acute wounds by varying the KJG concentration throughout the preparation procedure. It was shown that the hydrogel had water-holding capacity, biocompatibility, and antibacterial activity [199].

### 4.13. Guar Gum (GG)

Cyamopsis tetragonolobus is the source of GG, a plant-based polysaccharide with a high molecular weight. It consists of beta β-1→4-D-mannan linear chains and alpha (α)-(1→6)-linked galactose side chain units [200]. Guar gum is a hydrophilic polyhydroxy polysaccharide that is affordable. It is widely used in multifaceted and complex tissue engineering applications owing to its excellent rheological properties, strong biocompatibility, and biodegradability [201]. Indurkar et al. introduced a new, affordable bioink composed of gelatin and different concentrations of guar gum. This new biopolymer combination was developed to deliver bloom strengths, G′ and G″, that vary widely, and tan ∂ was calculated using G′ and G″ before the bloom test was carried out. These physical properties were linked to the bioink’s printability after the filament synthesis investigation. According to reports, the guar gum–gelatin bioinks could form filaments during extrusion, making them printable. Their bloom strength ranged from 480 to 750, and their tan ∂ ranged from 0.15 to 0.2 [202].

### 4.14. Pullulan

The molecular weight of pullulan, a non-ionic exopolysaccharide, varies between 10 and 400 kDa. It is composed of three glucose units joined by α-(1,4) glycosidic linkages, known as maltotriose units. Fermentation is a common method of producing Aureo-basidium pullulans. Cryphonectria parasitica and Tremella mesenterica are two micro-organisms that make pullulan. One polysaccharide that is derived from low-viscosity solutions is pullulan, which is very soluble in water. Pullulan could also be chemically changed to produce hydrogels by producing derivatives of methacrylate.

Qi et al. mixed pullulan altered with methacrylate moieties with poly(ethylene glycol) diacrylate (PEGDA) to create a hydrogel for cartilage tissue engineering [203]. Pullulan methacrylate and PEGDA were coupled by Giustina et al. to enable multiscale light-assisted 3D printing methods, including stereolithography and two-photon lithography [204]. Furthermore, pullulan was functionalized to offer active sites for cell attachment using the high molecular weight glycoprotein fibronectin. Because mesenchymal stem cells were able to attach to the extra fibronectin, MSC cell viability rose above 70% for a variety of time periods after cell seeding.

## 5. Stimuli-Responsive Polysaccharide-Based Hydrogels

### 5.1. pH-Responsive Polysaccharide Hydrogels

The development of innovative drug delivery systems (DDSs) and other cutting-edge technologies for medical applications has been greatly aided by the ability to modify the properties of natural polymer-based hydrogels in response to external stimuli such as pH, temperature, light, ultrasonic, enzyme, glucose, magnetic, redox, and electric (Figure 3) [205].

In biomedical applications, stimuli-responsive hydrogels containing labile connections inside the hydrogel matrix have garnered a lot of interest. Schiff bases—imine, hydrazine, and acyl hydrazone bonds—have been widely employed in the production of pH-sensitive wound dressings. These bonds have reversible characteristics that allow the creation of self-healing materials. They are readily generated by the reaction of a carbonyl group, an aldehyde, or a ketone with a primary amine, a hydrazine, or a hydrazide, respectively (Figure 4). Furthermore, it was demonstrated that hydrogels with reversible Schiff base connections had greater mechanical strength than those that are physically cross-linked [206].

Often obtained by oxidizing natural polymers, polysaccharides with aldehyde functionalities have been used extensively in the production of hydrogels with acid-sensitive Schiff base cross-links. Some of these polymers include carboxymethyl chitosan and oxidized hyaluronic acid (OHA), which are cross-linked to create a hydrogel for diabetic wound dressings [208]. For application in chronic wound healing, 4-arm poly(ethylene glycol) (PEG), an aldehyde-functionalized synthetic polymer with a complicated architecture, has also been cross-linked with natural polysaccharides such as carboxymethyl chitosan [209].

Zhang et al. also described pH-responsive nanocomposite hydrogels made of chitosan, oxidized hydroxypropyl cellulose (HPC), and octa (γ-chloroammoniumpropyl) silsesquioxane (OCAPS), a water-soluble POSS [210]. While OCAPS was attached to HPC via imine bonds, the aldehyde groups of HPC reacted with the amine groups of chitosan to generate the acid-labile Schiff base connections. In another instance, Khan et al. created pH-responsive multi-composite hydrogels using the natural polymers carrageenan and arabinoxylan in combination with reduced graphene oxide (rGO) and cross-linked with tetraethyl orthosilicate (TEOS) via the production of hydrogen bonds [211]. The hydrogels showed pH-dependent swelling, peaking at neutral pH 7 and seeing a reduction in swelling in both acidic and alkaline environments. Furthermore, the composite hydrogel showed a greater Young’s modulus than real human skin, as indicated by the stress–strain curves, and the tensile strength rose as the concentration of rGO in the material increased. Consequently, compared to actual human skin, the finished composite material had superior mechanical qualities.

In order to reduce inflammation and hasten the healing process, functional wound dressings that combine pH-responsive behavior with potent antibacterial moieties have lately become a popular option. Using oxidized dextran (OD) and cationic quaternized chitosan, Hoque and colleagues demonstrated the in situ synthesis of an antibacterial and bioadhesive hydrogel in 2017 [212]. Unlike CS, which underwent quaternization using GTMAC to produce the antibacterial polymer N-(2-hydroxypropyl)-3-trimethylammonium chitosan chloride (HTCC) (44% degree of quaternization), OD was produced by reacting the hydroxyl groups of dextran with sodium periodate at a 51 ± 1% degree of oxidation. Acid-labile Schiff base cross-links were produced by reacting the free amino groups of HTCC with the aldehyde groups of OD to make OD-HTCC hydrogels. Based on the HTCC concentration (2, 3, 4, and 5 *w*/*w*%), the hydrogels formed in 10–60 s. Faster gelation periods, extremely porous structures, and a progressive drop in the storage modulus (G′) were the results of lower polymer concentrations.

Additionally, the hydrogels demonstrated strong bioadhesion, which is typically a drawback of the majority of conventional bioadhesives, with high adhesive stresses in the range of 4.05 to 7.4 kPa. In related research, OHA was used to create acid-sensitive, antimicrobial hydrogels by utilizing adipic acid dihydrazide modified hyaluronic acid (HA-ADH) to cross-link its aldehyde groups and create acylhydrazone junctions. To give the hydrogels their antibacterial qualities, sisomicin sulfate or N(O)-protected QCS were chemically bonded onto the aldehyde groups via imine linkages [213,214]. More recently, QCS and OHA were cross-linked using imine bonds to create composite hydrogels [206]. The hydrogels were loaded with poly(3, 4-ethylenedioxythiophene):poly(styrene sulfonate) (PEDOT:PSS) as a photothermal agent, berberine (BBH), a natural antibacterial agent, and the epidermal growth factor (EGF). The hydrogels then released their cargo at the acidic wound area. When exposed to near-infrared (NIR) light, PEDOT’s presence allowed for a controlled increase in the antibacterial action.

Hydrogels that combine pH-responsive qualities with a self-healing action that preserves the dressing’s integrity when applied to deformable body parts are also of great interest in the creation of “smart” wound dressings. For joint skin wound dressings, pH-responsive, self-healing hydrogels based on QCS and aldehyde functionalized Pluronic F127 (PF127-CHO) micelles were made [215]. While PF127-CHO was made by reacting PF127 with 4-hydroxybenzaldehyde after the hydroxyl groups of PF127 were mesylated, QCS was created utilizing GTMAC. The hydrogels were created through the creation of imine bonds between the amine groups of QCS and the PF127-CHO curcumin-loaded spherical micelles, which were adorned with functional aldehyde groups at the outer shell. The Pluronic micelle physical cross-linking contacts and the Schiff base connections between the micelles and the QCS were responsible for the hydrogels’ exceptional self-healing capabilities.

In a different study, hydrogels that are dual-cross-linked, injectable, sticky, pH-responsive, and inspired by mussels were created to treat chronic diabetic wounds [216]. Dopamine-conjugated oxidized dextran (OD-DA) and the antibacterial polymer HTCC reacted to generate the “smart” hydrogels by the creation of acid-labile imine linkages between the free amine groups of HTCC and the aldehyde groups of dextran. Following gelation, catechol–catechol junctions were created within the hydrogel matrix by oxidizing dopamine with sodium periodate, resulting in a double-cross-linked network. Without changing the hydrogels’ viscoelastic characteristics, the addition of silver nanoparticles (AgNPs) and the pro-angiogenic medication deferoxamine (DFO) to its pores improved its antibacterial qualities and gave it an angiogenic character. The AgNPs and DFO were released at the wound site as a result of the cleavage of the Schiff base bonds under acidic circumstances, as those present in an infected diabetic lesion (pH 4–6). Additionally, the hydrogels’ capacity for self-healing was examined. Under an applied stress of 200%, complete gel degradation was seen, resulting in G″ > G′ and a transition to the liquid state. However, upon reducing the tension, the hydrogel recovered, and its mechanical characteristics were restored, demonstrating the hydrogel’s dynamic nature.

Lastly, Li and colleagues demonstrated a pH-responsive, self-healing hydrogel for diabetic wound healing that is based on N-carboxyethyl chitosan (N-chitosan) and OHA cross-linked with adipic acid dihydrazide (ADH) via the production of acylhydrazone and imine bonds, respectively [217]. In alkaline conditions, chitosan and acrylic acid reacted to produce N-chitosan, whereas HA was oxidized with sodium periodate to produce OHA. In less than 30 s after the diluted solutions of OHA, N-CECS, and ADH were mixed, stable hydrogels were created. The progressive elimination of a hole in the middle of the hydrogel after three hours and the measurement of the hydrogels’ storage and loss moduli (G′, G″) during many sol-to-gel cycles provided visible proof of the hydrogels’ self-healing capabilities.

By encouraging the re-epithelization of the injured skin and enhancing angiogenesis, insulin was injected into the hydrogels’ porous structure to speed up the healing process. Because of the breakdown of the acid-labile acylhydrazone and imine linkages, insulin was released more quickly at pH 6.5 than under neutral circumstances [217].

### 5.2. Redox-Responsive Polysaccharide Hydrogels

Dressings that react to a redox stimulation are greatly desired because the redox potential in an infected and injured skin area varies as the healing process progresses. Disulfide bonds are frequently utilized in the creation of redox-responsive materials since they are known to form reversibly under both oxidative and reductive circumstances. By creating amide bonds between the amino groups of aminoethyl disulfide (AED) and the carboxylic acid groups of HA, Gao et al. suggested a redox-responsive hydrogel made of HA cross-linked with a GSH-sensitive cross-linker [218]. The hydrogel was biocompatible and promoted fibroblast growth and proliferation for a period of seven days. However, when GSH was present, the hydrogel’s disulfide connections at the cross-link junctions were broken, which caused the hydrogel to gradually disintegrate. The morphological alterations of the GSH-sensitive hydrogels were suggested for application in detecting the healthy tissue throughout the healing process since diabetic foot ulcers had lower GSH levels than healthy tissues.

### 5.3. Photo-Responsive Polysaccharide Hydrogels

In wound healing, photo-responsive hydrogels are very appealing because they provide spatiotemporal control of the applied stimulus, which in turn affects the mechanical characteristics and release profiles of the hydrogels’ encapsulated active ingredients. A new light-responsive supramolecular hydrogel was created by Zhao and colleagues using azobenzene and β-cyclodextrin (β-CD) moieties grafted along hyaluronic acid chains as the host and guest [219]. Upon exposure to ultraviolet light, the well-known photo-responsive chemical azobenzene undergoes a trans-to-cis transformation. While cis-azobenzene does not interact with β-cyclodextrin, trans-azobenzene has a high affinity for its hydrophobic cavity, which encourages the establishment of host–guest interactions. A stiff supramolecular hydrogel with G′ ~155 Pa was created by the spontaneous interaction of HA chains conjugated with cyclodextrin (HA-CD) and HA chains conjugated with trans-azobenzene (HA-Azo). G′ dropped to about 144 Pa upon UV irradiation, indicating the reduction of the cross-link density of the hydrogel due to the isomerization of azobenzene from the trans to the cis isomer, whereas G′ was only gradually recovered in the dark or under visible light irradiation, which encourages the back isomerization of azobenzene to the trans conformation. Finally, EGF’s photo-controlled release profile was made possible by the hydrogel cavities’ capacity to encapsulate it, which promoted angiogenesis and wound healing.

## 6. Dual Responsive Hydrogels

### 6.1. pH- and Temperature-Responsive Polysaccharide Hydrogels

Reversible gels can occur above or below the critical solution temperature of thermoresponsive polymers. Specifically, because of their in situ gelling capabilities, polysaccharide-based polymers with a lower critical solution temperature near body temperature are highly appealing for wound healing applications [220]. However, hydrogels that have dual responsive characteristics—that is, the ability to recognize two orthogonal stimuli either independently or in concert—may display dynamic behavior that resembles the intricacy of real systems.

For diabetic wound healing, Wang and colleagues created pH- and thermoresponsive hydrogels with UV-shielding capabilities [221]. The hydrogels were created by oxidizing the natural polymer pullulan to produce aldehyde pullulan (APu) and then using Schiff base bond formation to cross-link it with Pluronic F127-grafted polyethylenimine (PEI).

At 4, 25, and 37 °C, the hydrogels’ temperature-responsive gelation was evaluated for varying APu concentrations; at temperatures higher than 25 °C, a sol-to-gel transition was noted [221]. Through electrostatic interactions, exosomes generated from adipose mesenchymal stem cells (ADSCs) were loaded into the hydrogel. When the acid-labile imine bonds were broken at pH 5.5, a more noticeable exosome release from the hydrogel was seen, which aided in the efficient healing of wounds.

Ma et al. created a different pH- and thermoresponsive multi-composite hydrogel based on hydroxypropyl chitin (HPCH) coupled with ferric ions and tannic acid (TA) [222]. Propylene oxide was grafted onto chitin chains in an alkaline environment to create HPCH, and the components were simply mixed at the appropriate concentrations to create the multi-composite HPCH/TA/Fe hydrogels. Instead of forming a homogenous network, mixtures of TA at concentrations more than 4 mg/mL and HPCH at 2 weight percent produced coacervates. Additionally, it was noted that the hydrogels lacked thermoresponsive qualities at TA concentrations between 0.5 and 3 mg/mL. Fe ions that interacted with TA while maintaining the hydrogel’s reversible thermoresponsive activity were employed by the authors to get around this restriction. For the HPCH/TA/Fe mixture, oscillatory rheology measurements revealed that the sol-to-gel transition occurred at about 18 °C and was finished at 37 °C. Furthermore, the release profile of TA from the composite hydrogels was examined at various pH levels. Because TA’s pyrogallol/catechol groups deprotonate when exposed to acidic circumstances (pH 3.0), the release was shown to increase while decreasing in neutral and alkaline settings. These groups form a strong complex with the Fe ions [45].

### 6.2. pH- and ROS-Responsive Polysaccharide Hydrogels

For the treatment of infected wounds, dual pH-and ROS-responsive hydrogels are attractive options because of the acidic environment and the elevated ROS content in these wounds [223]. In order to bind the boronic acid functions to the alginate backbone, Hu et al. recently presented a dual pH- and ROS-responsive hydrogel based on sodium alginate modified with 3-aminophenyl boronic acid (ALG-BA) [224]. An amphiphilic HA-CHOL polymer was produced by esterifying HA with cholesterol (CHOL) monomers. ALG-BA created a hydrogel with dynamic boronic ester linkages that are responsive to pH and ROS under alkaline (pH 8–9) conditions. The hydrophobic, anti-inflammatory medication naproxen was loaded into spherical nanoparticles made of the amphiphilic polymer HA-CHOL. Amikacin, an antibacterial medication, and the drug-loaded HA-CHOL micelles were confined inside the hydrogel matrix. The breaking of the boronic ester linkages caused the hydrogel to dissociate at high ROS concentrations and acidic conditions, which increased the release of the antibacterial and anti-inflammatory medications [224].

### 6.3. pH- and Electro-Responsive Polysaccharide Hydrogels

In order to treat chronic conditions that require regular injections or exact dosages of medication, electrical stimuli have recently become an appealing alternative. Electrical fields may be applied easily and precisely, making them a useful tool for wireless implants and medication delivery systems. Conducting polymers are electro-responsive materials that have natural antibacterial qualities and superior biocompatibility, among other advantages. Based on a chitosan-g-polyaniline (CP) graft copolymer, Qu et al. created a “smart” dual pH- and electro-responsive polysaccharide hydrogel in this regard [225]. A hydrogel with pH-labile imine linkages was created when the CP polymer and oxidized dextran (OD) interacted. This hydrogel demonstrated remarkable responsiveness to changes in solution pH as well as to an applied electric field. Amoxicillin, a hydrophilic negatively charged medication, exhibited a pH-dependent release profile when encapsulated in the hydrogel.

In contrast, both hydrophilic and hydrophobic drugs exhibited voltage-dependent release profiles when an external electric field was applied at an applied voltage ranging from 0 to 3 V. Similarly, this team used N-carboxyethyl chitosan (N-CECS) and an oxidized hyaluronic acid-graft-aniline tetramer, cross-linked via Schiff base linkages, to create multipurpose wound dressings that responded to pH and electrolytes and had increased granulation tissue thickness, collagen disposition, and angiogenesis for better skin regeneration [226].

## 7. 3D Printing Hydrogels for Tissue Engineering and Drug Delivery

### 7.1. Alginate and Its Derivates for Biomedical Use

Only a few polysaccharides exhibit thermal stability regarding melt strength or viscosity during printing, even though many polysaccharides have been tested for their potential printability. The goal is to combine them to create compounds with complementary properties, such as mechanically stable printable polysaccharides. A 3D-printable electroactive hydrogel of oxidized alginate gelatin developed by Distler et al. enhances cytocompatibility and cell–material interaction [227]. Hydrogel viscosity is reduced when ADA-GEL hydrogel precursors containing Py are produced. Direct extrusion printing was used to create gelatin-content oxidized alginate-gelatin polypyrrole (ADA-GEL-PPy) with specific rheological characteristics for 3D printing scaffolds. ADA-GEL-PPy hydrogels could be made with more electrical conductivity when PPy was included than pure ADA-GEL. Because 3D printing introduces porosity, it can improve the efficiency of seeding cells in the depth of hydrogel scaffolds. The discovery that PPy creation inside the hydrogel concurrently enhances electrical conductivity and hydrogel stiffness underlines the significance of comprehending the impact of CP alteration on many hydrogel characteristics simultaneously, which may eventually affect cell–material interaction. These characteristics and 3D printability may improve the effectiveness of existing cartilage tissue manufacturing techniques. These hydrogels are potential 3D-printable electroactive hydrogel matrices for research on electrical cell stimulation-aided tissue culturing. They may be used in sophisticated matrix-associated chondrocyte implantation applications or as drug-release electrode coatings.

Gel/PCL/PDA and core/shell fiber scaffolds were created by coating PDA and polycaprolactone (PCL) on drug-filled alginate/gelatin scaffolds that were 3D printed. NIR irradiation was used to provide on-demand medication delivery from the scaffolds. Long-lasting and maintained drug release may be achieved using PCL coatings by reducing the free diffusion of medicines from scaffolds. Chemo-photothermal treatment could completely eradicate malignancies and even effectively stop tumor development. The scaffolds made of core/shell fibers may also aid in the healing of wounds. The manufactured Gel/PCL/PDA core/shell fiber scaffolds offered the framework for targeted cancer treatment and fostering tissue regeneration, such as wound repair [228].

Dutta et al. combined CNC-incorporated gelatin and alginate to develop a bioactive and biodegradable hydrogel that may be used as a bioink for tissue engineering applications in 3D printing [229]. The composite hydrogel boosted cross-linking in comparison with a pure polymer hydrogel. This propensity was exacerbated by the incorporation of cross-linking agents into the created scaffolds. The fact that the structure of the produced scaffolds remained unchanged after freeze-drying proved their durability. More contact between the polymer chains and the embedded CNCs accompanied this improved mechanical strength. The swelling potential of composite scaffolds was less than that of pure polymer scaffolds because of the higher tendency for cross-linking. The printed scaffolds’ biocompatibility was shown by their presence having no adverse effects on Human Bone Marrow Mesenchymal Stem Cells (hBMSCs). Compared to the control, the osteogenesis-related gene expression rose in printed composite scaffolds, suggesting a higher potential for osteogenesis.

Using coaxial 3D printing, magnetic hollow fiber scaffolds containing iron oxide and alginate nanoparticles were produced. Gels containing medications, proteins, and cells can be encapsulated in the hollow core of scaffolding fibers. The drug-loaded hydrogels may be driven out of the open ends of the hollow scaffold upon magnetic stimulation. This allows for on-demand release. To enable carefully regulated distribution over time, the hollow fiber may sever and operate as an obstacle to reducing the unregulated dispersion of loaded medications and proteins. The amount of alginate, the strength of the cross-linking, and the exposed ends of the hollow fibers can all influence the drugs released from scaffolds. The developed hollow fiber scaffolds can be utilized for tissue engineering and disease therapy platforms due to their excellent biocompatibility and capacity to release cells and drugs as needed [230].

Olate-Moya et al. made scaffolds using 3D printing and inks and introduced brand new bioconjugated nanocomposite hydrogels made of graphene oxide (GO) particles, gelatin (GEL), and alginate (ALG) cross-linked [231]. Due to better thixotropic behavior brought about by adding GO to the ink formulation, printability was increased. Furthermore, scaffolds with improved form accuracy and resolution were created compared to ink lacking GO. Since GO is a naturally occurring liquid crystalline material, anisotropic fibers with engaging projections are present in ACG/GO1 threads for tissue engineering applications requiring cellular alignment. Using hADMSCs and GO-containing samples in 3D-printed bioconjugated scaffolds proved cytocompatible, and the scaffolds allowed for remarkable cell growth, alignment, and dispersion (Figure 5). An ALG polymer matrix that has been bioconjugated is intrinsically chondroinductive for the formation of hADMSCs, according to immunostaining tests conducted after four weeks of culture in a nonchondrogenic medium. Due to the biocompatibility and bioactivity of hADMSCs, 3D-printed scaffolds based on bioconjugated nanocomposite may make cartilage tissue engineering feasible. Plant-based nano-fibrillated cellulose (NFC) and carboxymethyl cellulose (CMC) composite inks were created using extrusion-based 3D printing.

The inexpensive, environmentally friendly, biocompatible inks developed by NFC and CMC in this study are perfect for simulating the ECM’s components (such as glycoproteins) and physiological processes. NFC/CMC ink can be used to make cell-filled materials and for 3D bioprinting because it is biocompatible with cells at low concentrations. Dry scaffolds can be handled and sterilized easily as well. The general technique described here is appropriate for producing durable biobased patterns for the growth and development of various biological tissues since both inks and scaffolds are highly versatile. They are adding cell adhesion sites and cross-linkers to make their system even more adaptable. During the bioprinting process, a granular support bath was used to cross-link natural biomaterials for the first time. The Schiff base cross-linking between the aldehyde groups of OAlg and the amino groups of Gel-CDH allowed for gelation without cross-linking procedures. It was shown that the rising Gel-CDH was bad for printability but better for cell survival and proliferation (Figure 6) [232].

Zhang et al. have modified the characteristics of hydrogels for 3D printing using spherical colloidal lignin particles (CLPs) [233]. CNF, alginate, CLPs, and novel biomaterial ink have shown astounding 3D printing skills. Shear-thinning behavior, an essential property for 3D printing biomaterial inks, was unaffected by an equivalent number of CLPs to dry CNF up to 25%. However, the end consequence was a higher printing resolution. Thanks to the inclusion of CLPs, the biomaterial inks now possess beneficial antioxidant properties. The form integrity of the printed scaffolds was enhanced by the additional cross-linking sites that CLPs for the divalent ions additionally provided, which were also present in the cell growth medium. High swelling ratios in every scaffold demonstrated excellent water absorption and retention capabilities. In addition, independent of the CLP concentration, HepG2 consistently grew in all the scaffolds that had been developed, demonstrating their high biocompatibility (Figure 7). The results suggested potential use for CLP-containing scaffolds in soft tissue engineering and regenerative medicine.

Gutierrez et al. created a unique technique for 3D-printing antimicrobial alginate/copper composites that are efficient against *E. coli* and *S. aureus* strains [234]. The study examined two cross-linking procedures using calcium ions first, copper ions and ion exchange second (method A), and just copper ions third (method B). A sodium borohydride solution was gradually infused into the hydrogels to create copper nanoparticle-infused alginate. Then, *E. coli* and *S. aureus* bacteria were successfully eliminated utilizing antimicrobial alginate/copper scaffolds created using Method A for 3D printing. The addition of BC nanofibrils greatly enhanced the printability of the alginate inks. It allowed for the creation of new antibacterial composite hydrogels with extended stability in dimension when exposed to CaCl2 solutions. Thanks to their discoveries in tissue engineering and regenerative medicine, a straightforward method for creating a particular class of 3D-printed antibacterial chemicals based on alginate is now accessible.

For extrusion-based 3D printing, novel T-CNF/SA hydrogel architectures were created. They were produced by printing calcium chloride at a concentration of 3% before partially cross-linking the hydrogel, depending on the weight of dried alginate. Hydrogel composed of 50% T-CNF and 50% alginate hydrogel (CNF50) duplicates the digital item with the highest printability and quality. T-CNF/SA scaffolds outperformed pure SA and pure T-CNF in terms of mechanical qualities, performing better at 50% compressive strain without breaking. Thus, mineralized T-CNF/SA scaffolds created by 3D printing have a potential application in bone tissue engineering [235]. Ionically cross-linked alginate-GO hydrogels have been shown by Valentin et al. to have improved chemo-mechanical stability [236].

Shear modulus is increased twice, inelastic deformation is reduced thrice, and fracture energy is increased ninefold in alginate-GO hydrogels compared to alginate-only hydrogels in shear, compression, and tension domains. Contrary to popular belief, alginate hydrogels can dissolve, but they do not degrade. In contrast, alginate-GO hydrogels keep their integrity when ionic cross-linkers are chelated and show increased stability due to hydrogen bonding. As a result, deleting or adding ionic cross-linkers allows for a more than 500-fold reversible tuning of the shear modulus. They show how to print hydrogel into several free-standing, dangling, and detachable 3D forms. These hydrogels can be employed for manipulating oil droplets and antifouling since they are still super oleophobic and structurally sound in a solution that resembles seawater.

Using cefazolin (CFZ) and rifampicin (RFP), Lee et al. developed a dual-drug-based three-dimensional scaffold, using PCL containing antibiotics and an exterior covering of alginate loaded with antibiotics [237]. These scaffolds based on dual drugs prevented the growth of *S. aureus* for a short time by inhibiting the production of biofilms. Dual-drug-based scaffolds were introduced to the LB broth during biofilm growth in order to investigate the impact of the RFP–alginate layer on the formation of *S. aureus* biofilms. The crystal violet assay was used to track the development of biofilm formation over the course of 48 h (Figure 8). Figure 8A shows the *S. aureus* biofilms stained with crystal violet on stainless steel coupons immersed in LB broth. Regardless of whether CFZ was present in the scaffolds, the RFP–alginate layer progressively decreased biofilm growth to about 50% over the course of 48 h. Together, these findings showed that RFP considerably decreased the production of *S. aureus* biofilms, which was in line with the absorbance measurement at 590 nm (Figure 8B). They could be identified by an early burst of RFP emission followed by a gradual and prolonged release of CFZ. This dual-drug scaffold offers fresh possibilities to enhance the management of osteomyelitis. [237].

### 7.2. Biomedical Applications of Chitosan Derivatives

Sui et al. used chitosan (CS) combined with an alkali/urea solution to create a hydrogel at 37 °C in a water bath [238]. The ‘green’ method avoids chemical cross-linking agents and different organic solvents. P-C-H ink was made by mixing lyophilized platelet-rich fibrin (L-PRF) and hydroxyapatite (HAP) with CS hydrogel.

A bone scaffold with the proper form, porosity, and connectivity was successfully created at a low temperature using 3D-printing technology. Four different scaffolds were made: P-C-H, 0.5% P-C-H, 1% P-C-H, and 2.5% P-C-H scaffolds; it was demonstrated that all four scaffolds exhibited considerable hydrophilicity and could degrade gradually. The mechanical characteristics of the C-H scaffold were the best; however, when the L-PRF rose, it gradually deteriorated. Consequently, 2.5% P-C-H might still fulfill the mechanical requirements of cancellous bone despite having the worst mechanical qualities. The exceptional antibacterial activity of the C-H scaffold was discovered by co-culturing the bacteria, which helped to justify the scaffold’s successful implantation. Although the scaffolds showed sufficient biocompatibility, the group with 2.5% P-C-H showed the highest level of bioactivity. As per the testing findings, the 2.5% P-C-H scaffold has remarkable biological activity to guarantee mechanical qualities. It applies to bone scaffolds [238].

Li et al. based their findings on tetrahedral framework nucleic acid (TFNA) recruitment directed by chitosan, TFNA-assisted cell proliferation, and chondrogenesis, creating a technique for cartilage regeneration using a hybrid PCL/CS hydrogel scaffold that includes synovial mesenchymal stem cells (SMSCs) [239]. Following articular cartilage (AC) abnormalities, the formation of the entire system simultaneously retarded the long-term progression of osteoarthritis. To encourage AC regeneration, this is the first research study integrating TFNA with SMSC-based scaffolds. This suggests a viable path for tissue engineering and AC regeneration [239].

Maturavongsadit et al. reported on developing a CS formulation containing CNCs as a bioink for 3D extrusion-based bioprinting. Extrusion-based 3D bioprinting works well with the CS/CNC bioinks, which also successfully induce osteogenic differentiation. The cross-linking procedure and composition of bioink formulations (Figure 9) were adjusted to enable effective printability [240].

The effects of CNCs and pre-osteoblast cells on the viscosity, yield stress, and recovery of the bioink formulations’ storage modulus were studied to forecast the results of 3D bioprinting. Adding CNCs and cells significantly enhanced the mechanical characteristics of 3D bioprinted carbon fiber scaffolds. All bioink formulations maintained a high density of 5 million cells/mL and were biocompatible. CS–CNC scaffolds significantly enhanced osteogenic differentiation, as seen by quicker alkaline phosphatase activity, calcium mineralization, and collagen formation during the early phases of extracellular matrix (ECM) development (Figure 10). They hope that this bioink will aid in creating biomimetic, bioprinted constructions with specific features that may be used to repair bony deformities efficiently.

Multifunctional mesh-type hydrogels were created and 3D printed by Alizadehgiashi et al. to enhance wound dressing applications for individualized wound therapy [241]. Chitosan meth acrylamide and cellulose nanocrystals were used to generate the host hydrogel. Physiologically active materials like proteins, small-molecule antibiotics, or silver nanoparticles were crammed into the hydrogel wound dressing’s distinctive filaments. The dressing’s host filament count was altered to affect how different drugs were passively released. The grid-type dressing’s 200 μm mesh size and 500 ± 100 μm diameter filaments are easily printed using 3D printing, allowing more flexibility in dressing architecture. Instead of Tegaderm controls, the physiological effects on different dressing formulations and release schedules were examined in vivo with multifunctional dressings. Hydrogels’ antibacterial and biocompatibility characteristics were assessed in vitro (Figure 11).

As early as six days following wound formation, animal models treated with hydrogel dressings exhibiting different Vascular Endothelial Growth Factor (VEGF) emission patterns demonstrated different physiological responses for producing granulation tissue and vascularization (Figure 12). Granulation tissue development was enhanced in the hydrogel-coated wound groups compared to the control groups. Variable amounts of vascular density were also detected at the wound site, depending on the hydrogel patterns employed. The hydrogel groups had significantly larger vascular densities than the control groups.

For their magnetic field sensitive self-healing characteristics, Choi et al. employed the oxidized hyaluronate (OHA), glycol chitosan (GC), and adipic acid dihydrazide (ADH) hydrogel and OHA/GC/ADH/SPION ferrogel. Creating dynamic tissue scaffolds via 3D printing was demonstrated [242]. The dimensional alterations of the 3D construct considerably impacted the chondrogenic growth of the ATDC5 cells that were contained and cultivated within it, contingent upon the intensity of the magnetic field. Using 3D and 4D printing techniques to build magnetic field-responsive structures for tissue engineering applications may prove to be a very promising application of this strategy.

A simple thermal/photo dual cure composite hydrogel design using soluble collagen and methacrylated HBC (MHBC) was presented by Liu et al. [243]. The MHBC and small molecule fish collagen (M/C) composite hydrogel showed rapid contraction and thermo-induced sol-gel transition for 3D cell culture. It also had the appropriate microstructure, adjustable mechanical properties, and biodegradability. The methacrylation influenced the degree of cytocompatibility in the MHBC and M/C ratios. Its printability was proven by the mild printing conditions, quick bioink gelation at 37 °C, and simple post-processing optimization of the ideal formulation (M/C 3/1). Since M/C composite hydrogel possesses the desired printability and cytocompatibility, it may be a viable bioink option for in situ 3D bioprinting.

Using the fused filament fabrication (FFF) 3D printing technique, Singh et al. showed that it is possible to create poly-lactic-acid (PLA) scaffolds reinforced with chitosan [244]. The following inferences may be made from the mechanical characterization carried out by statistical analysis: As the weight fraction of chitosan rises, tensile strength falls. This results from both the sliding of the chains on the chitosan and the discontinuities in the presence of chitosan particles introduced into the polymeric chains. Increasing infill density made it possible to raise tensile strength (TS). As chitosan’s weight percentage rises, the composite material’s density also rises, which boosts compressive strength. Flexural strength exhibits TS-like activity. However, the material exhibits a mix of brittle and ductile fractures due to the coexistence of tensile and compressive loads. The scaffoldings have FS values somewhat greater than TS and compressive strength (CS), making them particularly well-suited for dynamic movements. It is possible to accurately forecast the mechanical properties of TS, CS, and flexural strength (FS) using a single-parameter statistical analysis. This is demonstrated by the trials that used the optimal 3DP process parameter values. In contrast, multi-parametric optimization produced a composite desirability rating slightly above 50%.

A tri-solvent system consisting of dichloromethane (DCM), 2-butoxy ethanol (2-Bu), and dibutyl phthalate (DBP) was used to create a poly(L-lactide) (PLLA)-based ink. This improved the ink’s printability and the shape of the scaffolds during printing significantly more than a single-solvent ink that contained only DCM. PLLA matrix and chitin whiskers (CHWs) were present in various quantities in the composite inks made using this tri-solvent method. Following the direct ink writing (DIW) process, these inks were transformed into CHWs/PLLA (CP) porous composite scaffolds with the perfect porosity, pore size, and uniform shape. Compared to a pure PLLA scaffold, the CHWs improved the printed CP composite scaffold’s cell affinity, hydrophilicity, compressive performance, and osteogenic activity. However, it also refers to the variances in microstructure across composite scaffolds. The 40% CP scaffold was also shown to significantly improve the movement of macrophages between the M1 and M2 phenotype through the inflammatory response process compared to comparable scaffolds with lower mass concentrations of CHWs. This study demonstrates the ability of printed CP composite scaffolds to repair bone tissue [245]

According to Magli et al., Maleimide tetra-functionalized PEG (PEG-star-MA, available commercially) was used to functionalize chitosan and gelatin with methyl furan. Following that, the polymers produced were utilized to control the polymerization of composite polymers [246]. First, using the Diels–Alder process to create a network, the U87 cell line was used to evaluate the hydrogels’ viability for spheroid encapsulation and 3D bioprinting (Figure 13).

Seok et al. showed how to create ferrogel with the capacity for self-healing using polysaccharides and SPIONs [247]. For extrusion-based 3D printing, the gel could be used as an ink. A hydrogel was produced by combining GC and OHA solutions without excipient cross-linking agents. The concentration and ratio of polymers controlled the GC/OHA hydrogel’s properties. As the GC content rose, the hydrogel’s storage shear modulus decreased at the same total polymer concentration. In vitro, the stability of the hydrogel was excellent under physiological circumstances. When SPIONs were present, hydrogel changed into ferrogel, and the gel’s storage shear modulus decreased as the concentration of SPIONs increased. The GC/OHA/SPION ferrogel exhibited intriguing self-healing characteristics upon gel breakdown. Many opportunities for 4D printing in a magnetic field are presented by ferrogel extrusion printing, which makes it possible to produce 3D-printed objects in a range of shapes and sizes. Consequently, there are several potential uses for this polysaccharide-based self-healing ferrogel, such as administering medications and producing tissue for three-dimensional printing.

A magnetically activated biocompatible and biodegradable chitosan-based microswimmer with on-demand light-triggered medication release was created by Bozuyuk et al. [248]. Future research will thoroughly examine the optimization of the microswimmers in non-Newtonian bodily fluids because they are designed for swimming most effectively in the water. They created macromolecules of photosensitive meth acrylamide chitosan, then included SPIONs. The microswimmers were created using this material; they were 20 mm long and had an outer diameter of 6 mm. Under a 10 mT rotating magnetic field, they showed the microswimmers’ steering and actuation at various frequencies. Additionally, they demonstrated the biodegradation of the microswimmers utilizing a naturally occurring enzyme found in the human body without producing harmful in vitro degradation products. Last, the combination of OnDemand light-triggered medication release within the synthetic microswimmers was shown. The microsystem is promising for overcoming the difficulties involved in the active and regulated administration of medications to cure various ailments.

Using an innovative extrusion-based 3D printing technique, Intini et al. produced 3D chitosan biopolymeric scaffolds [249]. This technology may carefully control the 3D chitosan structures’ ultimate form and spatial arrangement. Two skin-associated human cell lines, Nhdf and HaCaT, exhibit exceptional cytocompatibility, toxicity, and biocompatibility with these scaffolds. These cell lines adhered to the 3D structures, colonized them, and multiplied quickly. The two cell lines co-cultured on 3D scaffolds with chitosan films at the base produced the most excellent results. Planted cells could not pass through the chitosan layer since it functioned as a barrier. Therefore, in addition to attaching and developing on the scaffolds, cells might grow up and through the scaffold framework. Chitosan film at the base of scaffolds improves their effectiveness during seeding because it prevents cells from leaving the pores created by the 3D printer. Because of this, the cells may remain where they are and have more time to adhere to the material. The chitosan scaffold is the surface proliferating cells that adhere to and expand, forming a continuous layer of linked cells. This is significant considering potential uses for skin integrity restoration. Furthermore, compared to wounds treated with a commercial product, a study conducted in vivo on diabetic rats using chitosan scaffolds to treat wounds promotes tissue regeneration and functioning, making them helpful in treating chronic cutaneous wounds.

### 7.3. Cellulose Applications in the Medical and Biomedical Fields

Dong et al. were successful in creating a novel hydrogel (Figure 14) with thermo- and UV-light-responsive characteristics with cellulose nanofibers and hyaluronic acid methacrylate (CN+HAMA) [250].

The microcellulose was thermosensitive, changing from a fluid to a gel at body temperature at low temperatures. The hydrogels can gel in place at temperatures higher than 30 °C and remain stable at lower temperatures, even below 0 °C, thanks to this reversible transformation. Furthermore, adding a UV-cross-linked HAMA polymer network may enhance the system’s temperature sensitivity. The CNs may also align directionally due to the shear stress created during extrusion. It produced guided soft-tissue mimics using hydrogels as a candidate ink because the cells on the printed scaffold formed in a specific direction. Natural and bio-extractive components comprise injectable, orientated, and dual-responsive hydrogels. They demonstrated strong biocompatibility (Figure 15), crucial for biofabrication. Finally, their work opens the study of highly directed biological tissue healing by creating a new printable ink with features including dual responsiveness, structural orientation, and biocompatibility [250].

A new composite with exceptional shape memory and tensile strength was created by combining Fe_3_O_4_ and cellulose nanofibers (CNFs) with poly-hydroxybutyrate/poly(ε-caprolactone) (PHB/PCL) mixtures as functional particles and reinforcing elements. Yue et al. evaluated how the mechanical properties and magneto-responsive shape memory performance of 3D-printed goods were affected by the inclusion of CNF and the printing conditions. Fe_3_O_4_ was added to the PHB/PCL mixture to improve tensile strength and reduce elongation at break. When the loading rate was 0.5% wt, CNFs had a major effect on the PHB/PCL mix’s ability to strengthen and hardness with 10% Fe_3_O_4_. Furthermore, experiments using 3D printing showed that the tensile strength and magneto-responsive shape recovery capabilities of the final products were significantly impacted by the addition of CNFs and modifications to the printing parameters. As functional evidence, a scaffold with exceptional load-carrying capacity was developed. Finding a new filament with remarkable tensile and magneto-responsive shape memory properties has expanded the range of materials available for 4D printing and raised the prospect of practical uses [251].

Mohan et al. used the market’s readily available NFC and CMC to make a unique ink known as NC3 [252]. The 1-weight percent NFC and 6-weight percent CMC in this ink exhibited the highest viscosity, shear moduli, and shear-thinning behavior compared to other evaluated ink formulations and pure CMC and NFC. Consequently, the printed structure possessed the maximum level of form fidelity. The printing strand spacing and dispensing pressure controlled the scaffolds’ microporosity. While forming interconnected micropores in each strand, freeze-drying preserved the print’s macroscopic appearance. By using dehydrothermal treatment, the scaffolds’ mechanical and moisture resistance properties were significantly improved, leading to physical cross-linking that preserved the prints’ shape. The prints’ surface hardness, compressive stress, and elastic modulus rose after the 120 °C treatment. Heat-treated biofluid scaffolds demonstrated long-term dimensional stability. The scaffolds were biocompatible, stimulated proliferation, and did not affect osteoblast cells derived from bone tissue.

Ji et al. described two new bioink materials that are cellulose-based macromers modified with norbornene [253]. The thiol: norbornene ratio (T: NB) and polymer content were adjusted to produce printable bioink formulations from cCMC. Printing cell-filled scaffolds and encasing cells was possible with all the ink formulations. These two cellulose-based macromers increase the number of possible bioinks. They might be changed to offer more enticing features with more usage and application.

MXenes, a novel family of 2D nanomaterials, have piqued researchers’ curiosity. Compared with other 2D nanomaterials, MXenes have demonstrated exceptional features, including photothermal conversion and electrical conductivity. To create intelligent fabrics and fibers constructed of MXenes/TOCNFs with exceptional sensitivity to diverse environmental stimuli, a unique approach was developed in this work. First, a simple mixing method was created (2,2,6,6-tetramethylpiperidine-1-oxylradi-cal)-mediated oxidized cellulose nanofibrils (TOCNFs) and Ti_3_C_2_ hybrid inks. Then, intelligent TOCNFs/Ti_3_C_2_ composite fibers and textiles with exceptional flexibility were produced utilizing a straightforward 3D-printing approach. The oxidized CNF hydrogel’s whole flow-deformation behavior was examined and studied by Sanandiya et al. [254]. The material’s properties are affected by its concentration and external stimuli such as chemicals, physical materials, or cell culture media. The hydrogel’s remarkable versatility in injection, scaffolding, and 3D bioprinting allows it to be employed as a delivery vehicle for cell encapsulation, enabling continuous release of paracrine hormones and tissue regeneration. MESCs and MCF-7 cells favorably received it.

To summarize, 3D scaffolds outperform traditional 2D cultures by providing a more realistic biomimetic environment. They offer more physiologically relevant research, as evidenced by the in vivo wound healing model. They are reflected in cost-effectiveness and quicker recovery. Compared to competing products, a product’s cost-effectiveness is determined by its capacity to be scaled up to an industrial level while maintaining a high reproducibility of essential properties like porosity. Starting materials for the process, such as biopolymers like alginate or collagen, are inexpensive. For further comparison Table 3 describe some preclinical printed scaffolds made of pol-ysaccharides.

## 8. Limitations

The capacity to precisely print high resolution constructs of various biomaterials, cells, and medicines for the creation of extremely complex 3D cardiac constructs is an advantage of the 3D bioprinting technique. A fully functional heart has not yet been fabricated, despite the tremendous advancements and expertise in 3D bioprinting in tissue engineering. The printing resolution is one of the main issues with 3D bioprinting. The optimal resolution must be like the cell size for the created replacement tissue to closely resemble the native tissue. It takes a multilayered tissue for clinical use. The creation of a regulated vascular network for cell survival is difficult [262]. Although 3D bioprinting has improved in recent years in terms of reaching structural complexity, soft-material 3D bioprinting is still in its infancy [263]. Another drawback of 3D bioprinting is the lack of a recognized and standardized technique to evaluate the precision and effectiveness of the models that are created, which leads to variability. The assessment of tissue behavior is limited because several of the biomaterials now in use do not accurately mimic the mechanical characteristics of the human heart [264]. The choice of scaffold and process parameter optimization present another challenge in 3D bioprinting. Additionally, when compared to natural cardiomyocytes, the current polymeric materials have poor mechanical strength and inappropriate conductivity. Degradability and biocompatibility of scaffolds, as well as cell migration and interaction within the scaffolds, continue to be significant challenges for CTE engineers. Three-dimensional bioprinting of heart tissue is still in its infancy, but once the problems are resolved, it will allow technology to be translated to customize pharmacological and therapeutic uses in the years to come.

3D printing, also known as additive manufacturing, has revolutionized various industries, including pharmaceuticals. However, despite its potential, there are several limitations and challenges associated with this technology, particularly in the context of drug delivery. The layer-by-layer deposition process inherent to 3D printing can lead to various defects and imperfections. These include the following: 1. voids: small gaps or voids can form between layers, compromising the structural integrity of the printed object. 2. incomplete fusion: inadequate bonding between layers can result in weak points within the structure. 3. surface roughness: the surface of 3D-printed objects can be rough, which may affect the release profile of drugs. 4. internal anisotropy: the mechanical properties of 3D-printed structures can vary depending on the direction of the layers, leading to anisotropy. The range of materials suitable for 3D printing in pharmaceuticals is limited. Not all materials can be processed using 3D printing techniques, and those that can may not always meet the required standards for drug delivery applications. Ensuring consistent quality and standardization of 3D-printed drug delivery systems is challenging. Variations in printing parameters, environmental conditions, and material properties can lead to inconsistencies in the final product. The regulatory landscape for 3D-printed pharmaceuticals is still evolving. Ensuring compliance with existing regulations and obtaining approval for new 3D-printed drug delivery systems can be a complex and time-consuming process. While 3D printing offers the advantage of customization, scaling up production to meet commercial demands remains a challenge. The speed of 3D printing is relatively slow compared to traditional manufacturing methods, which can limit its efficiency for large-scale production. The long-term effects of using 3D-printed drug delivery systems are not yet fully understood. Ensuring patient safety and addressing potential risks associated with the use of these systems is crucial. Despite these limitations, 3D printing holds great promise for the future of drug delivery. Ongoing research and technological advancements are expected to address many of these challenges, paving the way for more effective and personalized drug delivery solutions [265,266,267].

## 9. Conclusions and Future Prospective

Bioprinting is a highly intriguing scientific idea that could transform medical science in the future. The use of 3D printing in tissue engineering has the potential to dramatically transform our understanding of transplantation by removing all problems associated with organ waiting lists and immune compatibility and eventually producing a tissue substitute that is entirely customized and patient specific. Higher mechanical qualities, stimuli response, injectability, and other attributes have been conferred by the frequent combination of polysaccharides with other organic or inorganic components. Recently, there has been a lot of interest in the creation of stimuli-responsive hydrogels and their composites for wound healing applications. The most popular examples include hydrogels that are sensitive to pH and ROS, as well as materials that are sensitive to temperature, enzymes, and light. While lots of research has been carried out on different chemical, biochemical, and physical stimuli to cause physicochemical, mechanical, and morphological changes in polysaccharide-based hydrogels, there is a lot of interest in the development of polysaccharide composite hydrogels that can react to external stimuli like light irradiation, ultrasounds, and magnetic or electric fields.

While 3D bioprinted materials based on polysaccharide hydrogel have great potential for a range of biomedical applications, a number of obstacles must be overcome before their full potential can be achieved. These difficulties include restrictions related to fabrication methods and material characteristics. Notably, obtaining mechanical qualities similar to those of natural tissues is a major obstacle for structures based on polysaccharide hydrogels. Despite the fact that polysaccharide hydrogels have mechanical properties that can be adjusted, it is still difficult to optimize these properties to meet the needs of particular tissues. To improve the mechanical strength and stability, tactics including adding reinforcing agents, hybridizing with different polymers, or adjusting the cross-linking density must be investigated. Furthermore, the inherent rheological properties of polysaccharide hydrogel-based bioinks make it difficult to achieve high-resolution printing. However, optimizing printing parameters such as nozzle size, printing speed, and bioink viscosity is necessary to increase printability while preserving cell survival and structural fidelity. Furthermore, the advancement of innovative printing methods including volumetric additive manufacturing, nozzle-free printing, and multi-material printing may further increase resolution and flexibility. Furthermore, the development of innovative printing techniques like nozzle-free printing and multi-material printing may further improve resolution and adaptability. The significance of polysaccharide hydrogel-based 3D bioprinting, which balances stability and biodegradability to promote tissue regeneration while preserving structural integrity, was also made clear by this study. Therefore, regulating the rate of degradation and designing scaffolds that break down at a pace that corresponds with tissue regeneration continue to be important factors. Adding stimuli-responsive components or bioactive compounds to hydrogels may improve tissue integration and allow for regulated breakdown.

Researchers can employ sophisticated instruments beyond conventional laboratory techniques to tackle the many difficulties involved in creating and manufacturing hydrogels using 3D bioprinting. Computational simulation techniques, for instance, especially those augmented by artificial intelligence (AI), are proving to be effective allies. AI makes it easier to analyze different hydrogel properties and build complex models that connect structure, composition, process, and properties. This aids in understanding the rheological features of polysaccharides and their derivatives, as well as in forecasting and optimizing hydrogel qualities.

## Figures and Tables

**Figure 1 biomedicines-13-00731-f001:**
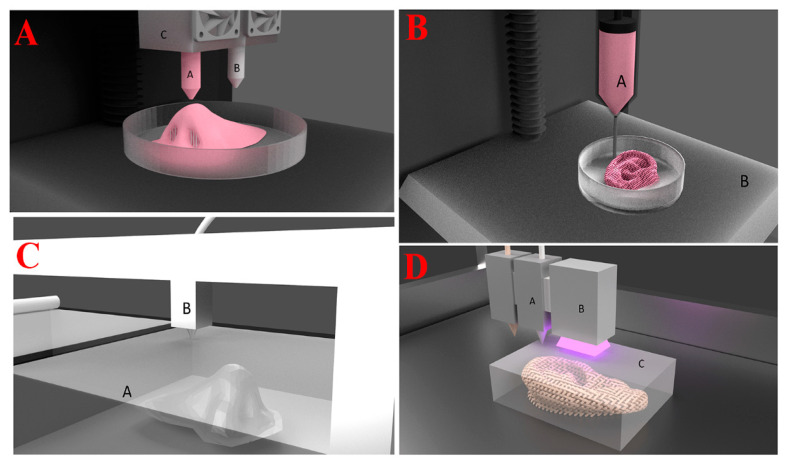
(**A**) Dual-head FDM 3D printer. (A) Building material; (B) supporting material; (C) print heads. (**B**) Extrusion-based bioprinting. (A) Bioink; (B) build platform. (**C**) Inkjet 3D printing. (A) Powdered bed; (B) binding liquid spraying nozzle. (**D**) PolyJet 3D printer. (A) Nozzle spraying photopolymer; (B) UV source; (C) supporting material. Adapted from [37] with permission.

**Figure 2 biomedicines-13-00731-f002:**
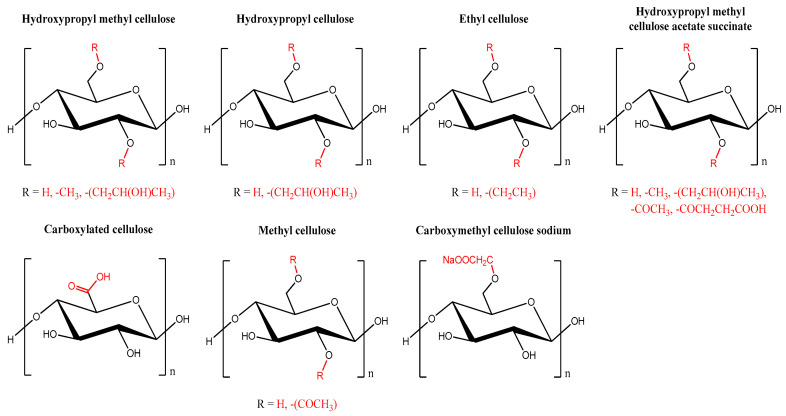
Structures of some common derivatives of cellulose. Adapted from [126] with permission.

**Figure 3 biomedicines-13-00731-f003:**
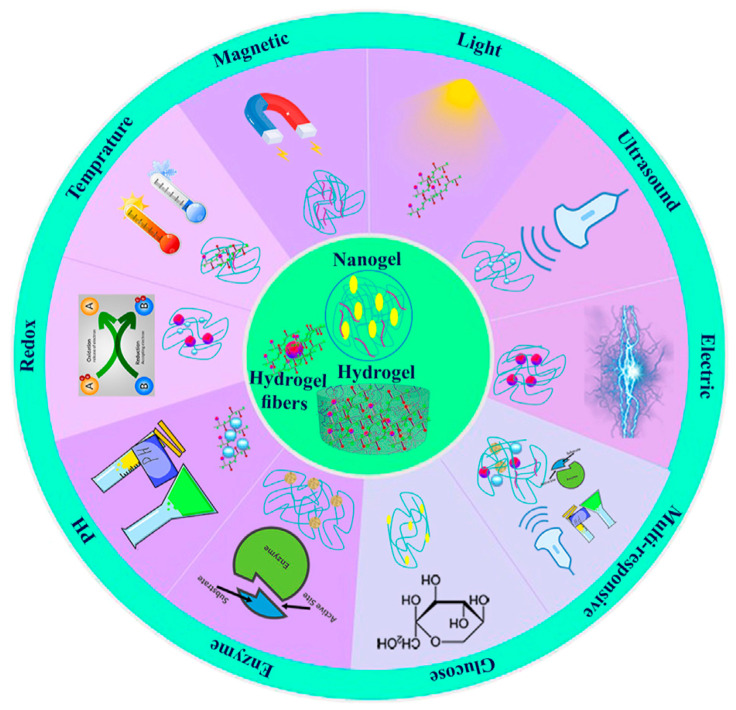
Classification of stimuli-responsive hydrogels. Adapted from [205] with permission.

**Figure 4 biomedicines-13-00731-f004:**
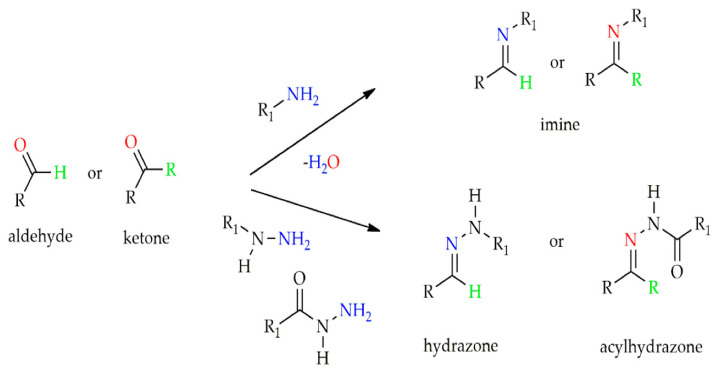
Formation of Schiff base linkages. Adapted from [207] with permission.

**Figure 5 biomedicines-13-00731-f005:**
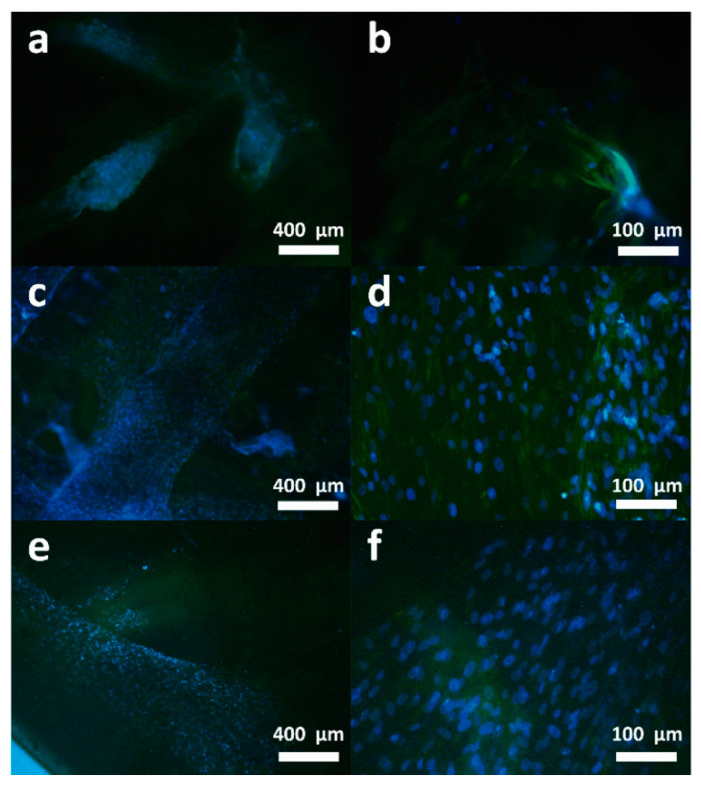
Fluorescence microscopy images of hADMSCs seeded on 3D printed scaffolds at day 7. Cytoskel-eton F-actin (green) and nuclei (blue) of hADMSCs are showed for (**a**,**b**) ACG, (**c**,**d**) ACG/GO0.1 and (**e**,**f**) ACG/GO1 scaffolds. Adapted from [231] with permission.

**Figure 6 biomedicines-13-00731-f006:**
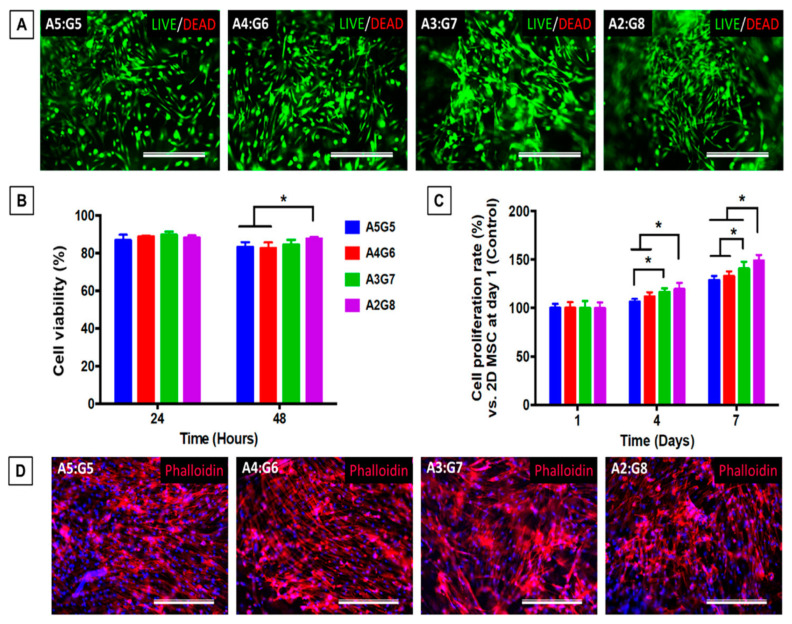
(**A**) Pictures of representative LIVE/DEAD staining at day 2 (scale bars = 400 μm), (**B**,**C**) cell viability by LIVE/DEAD assay (n = 5), cell proliferation rate by CCK-8 (n = 10), and (**D**) pictures of F-actin staining at day 2 (scale bars = 400 μm, * *p* < 0.05). Adapted from [232] with permission.

**Figure 7 biomedicines-13-00731-f007:**
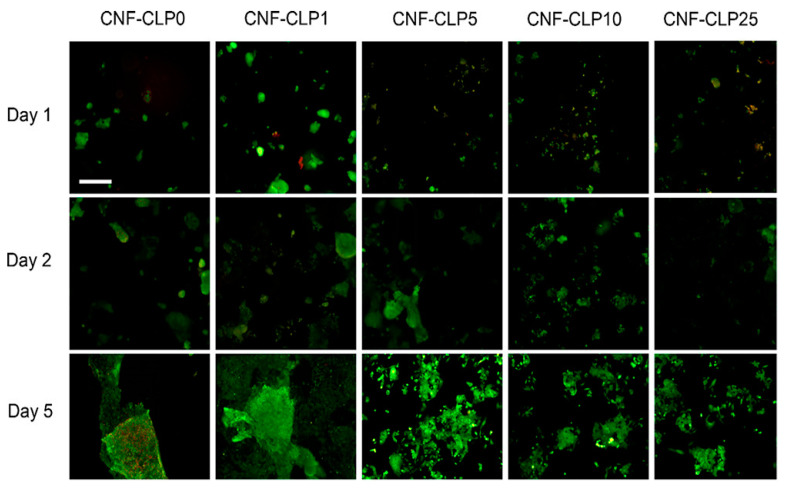
HepG2 cells seeded on the prepared scaffolds were shown in fluorescence microscopy pictures after one, two, and five days of incubation at progressively higher relative lignin concentrations to dry CNF (from 0 to 25%). One hundred μm scale bar. Adapted from [233] with permission.

**Figure 8 biomedicines-13-00731-f008:**
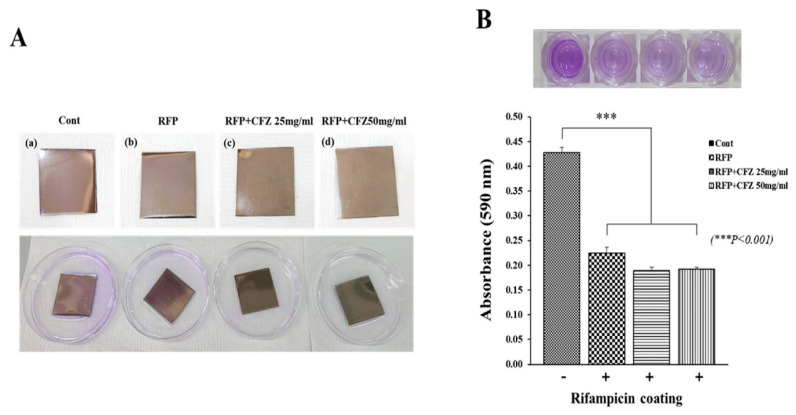
Staphylococcus aureus biofilm formation on stainless steel coupons: (**A**) Biofilm formation assays of *S. aureus* strains. The formed biofilm was stained with 0.1% crystal violet. (**B**) Quantification of *S. aureus* biofilm formation. Adapted from [237] with permission.

**Figure 9 biomedicines-13-00731-f009:**
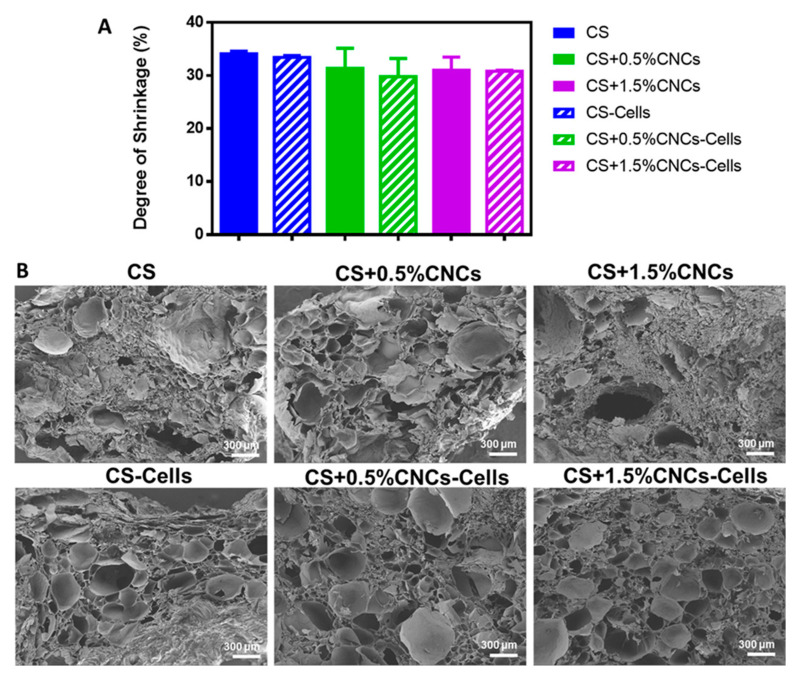
Physical characteristics of bioprinted scaffolds in three dimensions. (**A**) Following a 24 h incubation in DMEM at 37 °C, the degree of shrinkage of cylindrical bioprinted scaffolds (n = 4) made using various bioink formulations without and with MC3T3-E1 cells (5 million cells/mL) was measured. ANOVA with Tukey’s multiple comparison tests was used for the statistical analysis; there was no significant difference (*p* > 0.1). (**B**) Casted scaffold microstructure was observed during a 24 h incubation in DMEM at 37 °C. Adapted from [240] with permission.

**Figure 10 biomedicines-13-00731-f010:**
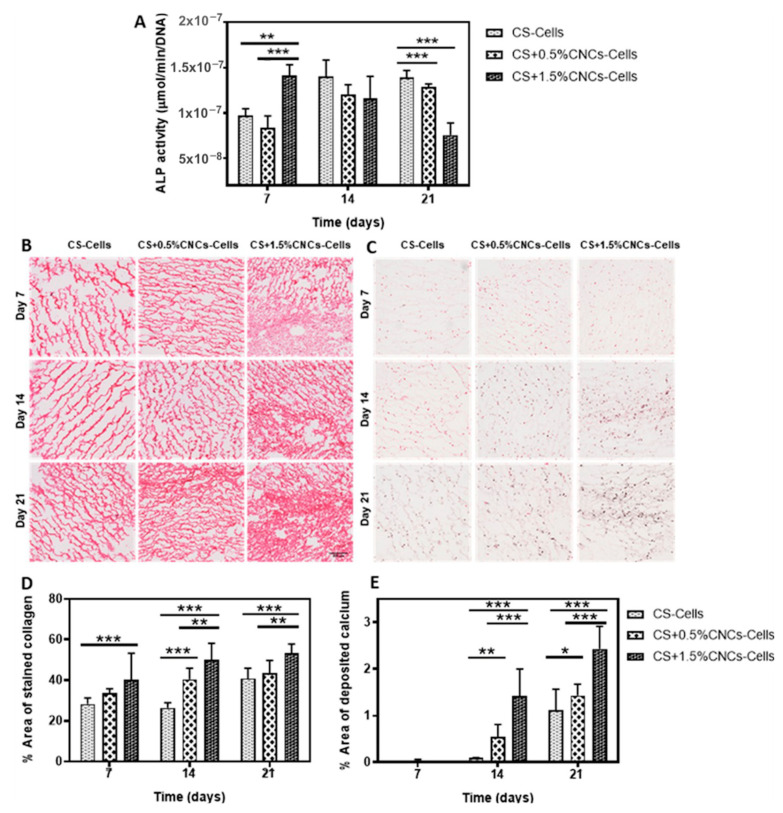
Three-dimensional bioprinted scaffolds used for in vitro osteogenesis tests (n = 5). (**A**) In osteogenic culture conditions, ALP activity was assessed at 7, 14, and 21 post-incubations using p-nitrophenyl phosphate (pNPP). (**B**) H&E staining of freshly generated extracellular matrix in bioprinted scaffolds tests at devaluated days 7, 14, and 21 after incubation in settings similar to osteogenic culture. The 200 μm scale bar is the same for all the pictures. (**C**) After incubation, Von Kossa staining of calcium mineralization in bioprinted scaffolds was obtained under osteogenic culture conditions on days 7, 14, and 21. The 200 μm scale bar is the same for all the pictures. (**D**) The percentage of freshly generated extracellular matrix (ECM) in bioprinted scaffolds at days 7, 14, and 21 after incubation under osteogenic culture conditions, as measured by ImageJ (https://imagej.net/ij/, accessed on 10 February 2025). (**E**) In bioprinted scaffolds, the percentage area of calcium mineralization was measured using ImageJ on days 7, 14, and 21 after incubation in osteogenic culture conditions. ANOVA with Tukey’s multiple comparisons tests; * *p* < 0.1, ** *p* < 0.05, and *** *p* < 0.01 are the statistical tests used. Adapted from [240] with permission.

**Figure 11 biomedicines-13-00731-f011:**
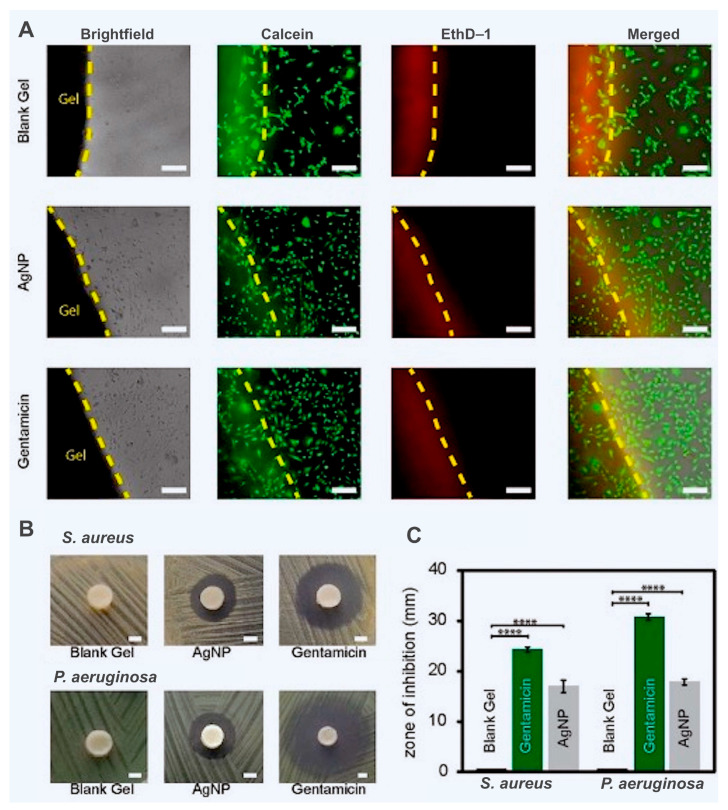
Hydrogels’ antimicrobial and biocompatible qualities. (**A**) Images from the LIVE/DEAD test of 3T3 cells cultured in hydrogels devoid of biologically active agents (blank gel) or loaded in six-well plates with specific biologically active agents during a 48 h incubation period in DMEM. Calcein AM was used to stain live cells (green), whereas EthD-1 was used to label dead cells (red). The dotted line shows the hydrogel’s edge. There is a 200 μm scale bar. (**B**) The Zone of Inhibition (ZOI) test was conducted on *S. aureus* (top) and *P. aeruginosa* (bottom) cultures in the presence of hydrogels that were loaded with gentamicin, AgNP, and antimicrobials (blank gels). (**C**) The scale bar is 5 mm. The average width of the ZOI for *S. aureus* and *P. aeruginosa* was assessed in the presence of hydrogels with and without antimicrobial drugs. The unpaired *t*-test yielded a **** *p* < 0.0001. Adapted from [241] with permission.

**Figure 12 biomedicines-13-00731-f012:**
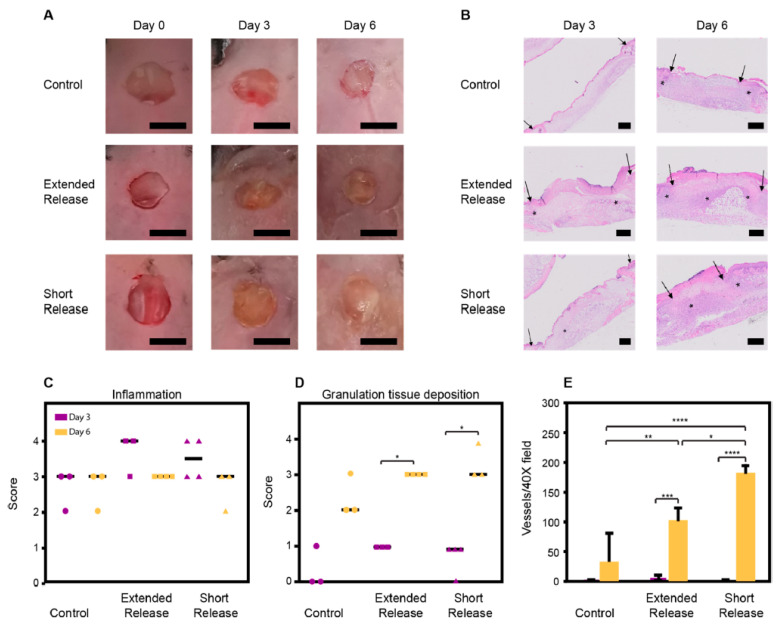
Examination of six days’ worth of wound healing in a mouse model. (**A**) Itemized images of lacerations on days 0, 3, and 6, with dressings changed on day 3. The scale bar is five millimeters. (**B**) Hematoxylin and eosin-stained representative slices of histopathological wound tissue samples were taken on days three and six. Asterisks indicate regions of granulation tissue deposition in the wound bed. In contrast, arrows indicate the borders of the skin at the wound’s margins. It has a 0.5 mm scale bar. (**C**) On days 3 (purple) and 6 (yellow), the degree of tissue inflammation was semiquantitatively evaluated by histology using a 5-point scale (0 = normal/none, 1 = minimum, 2 = mild, 3 = moderate, and 4 = noticeable). The median and scores are shown. (**D**) On day 3 (purple) and day 6 (yellow), granulation tissue deposition was measured by histology using the same scale as (**C**). The median and scores are shown. * Mann–Whitney tests yielded an exact *p*-value of 0.0286. (**E**) Histology was used to determine the tissue vascularization level. The average number of vascular profiles per five randomly placed, high-magnification (40×) regions inside the wound bed was counted in serial sections stained for CD31 on days three and six (yellow and purple, respectively). The two-way ANOVA test yielded adjusted *p* values of * *p* = 0.0181, ** *p* = 0.0054, *** *p* = 0.0006, and **** *p* < 0.0001. Adapted from [241] with permission.

**Figure 13 biomedicines-13-00731-f013:**
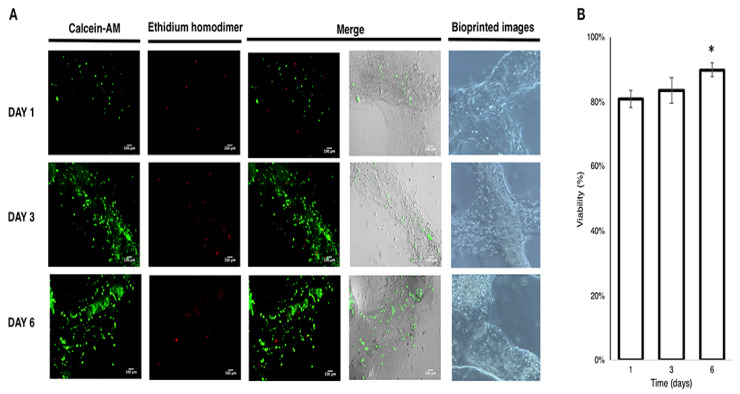
(**A**) LIVE/DEAD imaging of bioprinted U87 cells; (**B**) viability of bioprinted cells. (Mean ± SD One-way ANOVA, n = 3: * *p* < 0.005). Scale bars are 100 micrometer each. Adapted from [246] with permission.

**Figure 14 biomedicines-13-00731-f014:**
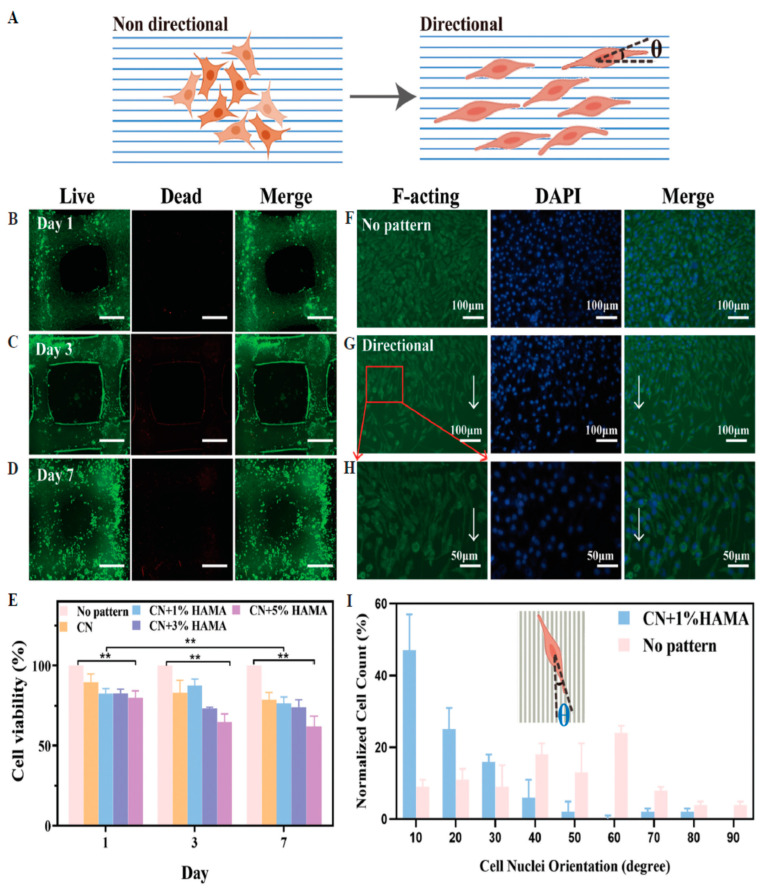
CN+1%HAMA hydrogels induce directional growth of L929 cells. (**A**) Schematic diagram of the growth process of cells on the surface of oriented fibers. (**B**) Day 1, (**C**) Day 3, and (**D**) Day 7 fluo-rescence images of printed scaffolds seeded with L929 cells with live/dead stain. The scale bars for images in (**B**–**D**) are 500 μm. (**E**) Cell viability comparison between various hydrogels on different days (Days 1, 3, and 7), measured by a CCK-8 assay. (**F**) The growth process of L929 cells seeded on culture dishes without treatment. (**G**) The growth process of cells seeded on printed CN+1%HAMA hydrogel scaffolds. (**H**) A zoomed-in view of the frame column section in Figure 15. (**I**) Orientation angle distribution of L929 cells on CN+1%HAMA hydrogel scaffolds after culturing for 7 days. Orientation angle dis-tribution of L929 cells on CN+1%HAMA hydrogel scaffolds after culturing for 7 days Adapted from [250] with permission.

**Figure 15 biomedicines-13-00731-f015:**
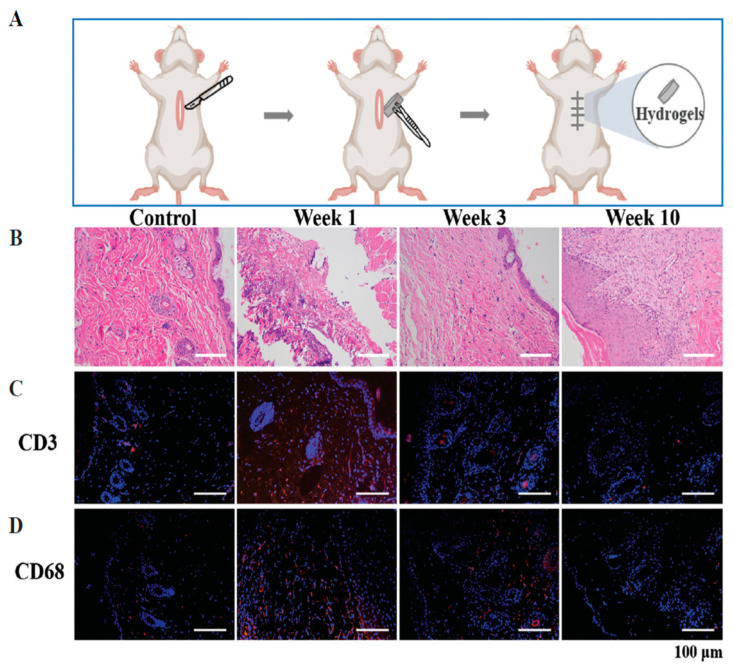
Kunming mouse models for the biosafety and characterization of CN+1%HAMA hydrogels. (**A**) Diagrammatic representation of hydrogel sample implantation in models of Kunming mice. After one, three, or ten weeks, hydrogels implanted subcutaneously were explanted with the surrounding tissue. Sections were stained red for CD3 (**C**) or CD68 (**D**) antigens after being exposed to H&E staining (**B**). Adapted from [250] with permission.

**Table 1 biomedicines-13-00731-t001:** Three-dimensional printing technologies, mechanism of action, advantages and disadvantages.

3D Printing Technologies	Mechanism of Action	Advantages	Disadvantages	References
Inkjet printing	Thin layer of active pharmaceutical ingredient with excipients on a solid platform is selectively bonded by sprayed formulations or binders in microdots	High resolution and precision, efficient, economical, high-speed manufacturing, multi-material printing	Nozzle clogging, low friability and hardness	[41,42,43,44]
Continuous inkjet	Pressurized continuous stream of droplets (50–80 μm) directed by electrostatic plates to solidify	High drop velocity covering longer distances, no nozzle clogging, faster output	High degree of wasted ink due to recirculation, limited availability of solvent-based inks	[41,42,43,44]
Drop on demand	Droplets (10–50 μm) through multiple nozzles—thermal or piezoelectric heads	Faster solidification, economical	Susceptible to nozzle clogging resulting in inaccurate jetting and dropping	[41,42,44,45]
Binder jetting	Polymer powder or other solid particles with liquid binder	Ability to produce porous constructs, multi-material printing, no support needed	Limited selection of materials, low structural integrity	[41,42,44,45]
Powder deposition technique	Sprayed drops from print heads are deposited as layer and laser beam sinters the powder layer to form solid structures	Average resolution and speed, no support needed, recyclable feed materials	Particle size of powder binding materials is critical, change in mechanical properties	[46]
Selective laser sintering	Melting and fusion of high-melting-point thermoplastic polymers and low-melting-point binding powder materials	High resolution (30 μm) and precision, porous structures, faster fabrication, no post-curing required	Low efficiency, expensive, significant wastage of powder materials, limited availability of active pharmaceutical ingredients and excipients suitable for the process	[47]
Fused deposition modelling	Materials extruded through a nozzle or orifice under controlled conditions to deposit in layer-by-layer fashion to form 3D object	Inexpensive, compact equipment, diverse, readily available, ecofriendly and non-contaminating raw materials, ability to create complex, innovative and customized dosage forms	Requirement of solvent, heat and cross-linking agents, difficulty to recycle printing materials, risk of drug and excipient degradation, slow printing speed, delamination due to temperature fluctuations	[41,48]
Injection molding	Molds created by auto-computer-aided design software in stereolithography file format and sliced into G-code	Creation of drug delivery systems with specific geometric shape and dimension, scalable, continuous manufacturing technique, no solvent requirement, mechanical anisotropy of structures is minimal	Limited design, relatively expensive technique	[5,41]
Stereolithography	Digital mirroring device utilizing laser beam to initiate photochemical reaction to transform liquid monomer into solid object	High accuracy and resolution, complex and customized drug delivery systems with desired release pattern, minimum drug decomposition, compact equipment, suitable for personalized dosage form development in clinical setting, minimum mechanical anisotropy	Potential toxicity, low drug loading, rinsing and post curing process is necessary, limited availability of biocompatible photopolymerizable polymers	[49,50]
Digital light processing	Laser beam projected through digital mirror device	High resolution, high-speed manufacturing	Toxicity, needs support	[49]
Continuous liquid interface production	Projecting ultraviolet light through oxygen permeable membrane	Fastest manufacturing speed, high precision	Probable toxicity, expensive	[49]
Semisolid extrusion system	Semisolid materials extruded through pressure-assisted microsyringe by compressed air	High-speed process, operation can be carried out at room temperature, suitable for thermolabile drugs, high drug loading, cost effective for bulk production	Specific rheological characteristics required for starting material, pseudoplastic and cross-linking polymers preferred, chance of nozzle clogging, resolution limited by nozzle size, drying step is necessary, low resolution, slow production speed, low mechanical strength and durability	[38,51]

**Table 2 biomedicines-13-00731-t002:** Comparison of different methods.

Designation Additive Manufacturing Process	Technologies	Medical Use	Pros	Cons
Vat photo-polymerization	Stereolithography (SLA) Digital light processing (DLP)	Bone, dental models, dental implant guides, hearing aids	High resolution and accuracy Complex parts Decent surface finish Flexible printing setup	Lacking in strength and durability They are still affected by UV light after printing Not for heavy use
Material jetting	MultiJet modelling (MJM)	Medical models, dental casts, dental implant guides	High accuracy Low waste of materials Multiple material parts and colors in one process	Requires support material Only polymers and waxes are supported
Binder jetting	Powder bed and inkjet head 3D printing (PDIH) Plaster-based 3D printing (PP)	Color models, especially color coding of anatomy	Range of colors Multiple materials supported Faster Different binder powder combinations	Not always suitable for structural parts Cleaning the 3D-printing result takes time and increases the time required for the procedure
Material extrusion	Fused deposition modeling (FDM) Fused filament fabrication (FFF)	Medical instruments and devices, rapid prototyping exoskeletons	Inexpensive process Widespread ABS plastic supported	Dependence of quality on the nozzle radius Low accuracy Low speed Contact pressure needed to increase quality
Powder bed fusion	Selective laser sintering (SLS) Direct metal laser sintering (DMLS) Selective heat sintering (SHS) Selective laser melting (SLM) Electron beam melting (EBM)	Models that require a lattice, medical devices such as implants, and fixations	Inexpensive Small technology Extensive range of material options	Low speed Limited sizes Dependence on powder grain size

**Table 3 biomedicines-13-00731-t003:** Type of printing and use of printed scaffolds made of polysaccharides.

Materials	Printing Type	Response to Stimuli/Biomedical Application	References
Agarose/acrylamide	Situ polymerizing	Temperature/human ear or nose printing	[255]
Alginate glycerin hydrogel	Microfluidic coaxial extrusion	PH/skin dressing	[256]
Chitosan	Plasma polymerization	PH/surface modification	[257]
Chitosan/methacrylated alginate	Extrusion bioprinter	Voltage/vascular stents	[258]
Hyaluronic acid/polycaprolactone	Laser sinter	Tension/tracheobronchial splint	[259]
Hyaluronic acid/polylactide	Fused deposition modeling	Temperature/orthopedic implant	[260]
Sodium alginate/agarose/*N*, *N*′-methylene bis (acrylamide)	Laser-machining and screen printing	Temperature/patch	[261]

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
