# Peer review of "3D Printing of Hydrogel Polysaccharides for Biomedical Applications: A Review"

_biomedicines, 2025, doi:10.3390/biomedicines13030731_

Round 1

Reviewer 1 Report (Previous Reviewer 2)

Comments and Suggestions for Authors

see attached file

Author Response

Reviewer 1

  1. The scientific background and the aim of the review are presented unsatisfactorily. For the reasons below, the relevance of this current review is not obvious and the interest to the scientific community is very low.

1.2 Recently, much attention has been paid to 3D bioprinting as one of the inspiring technologies in biomedicine and a wide range of review articles have been published in the last 3 years. Below is a list of some of them. Unfortunately, none of them are mentioned by the authors in the Introduction. As a result, it remains unclear which gap in knowledge was identified by the authors as important to fill. It is strongly recommended to highlight those points that distinguish the study from the analysis performed by other authors earlier.

Damiri F, et al. Recent advances in 3D bioprinted polysaccharide hydrogels for biomedical applications: A comprehensive review. Carbohydr Polym. 2025 Jan 15;348(Pt B):122845. doi: 10.1016/j.carbpol.2024.122845

Shokrani H, et al. Polysaccharide-based biomaterials in a journey from 3D to 4D printing. Bioeng Transl Med. 2023 Mar 22;8(4):e10503. doi: 10.1002/btm2.10503

Yao Z, et al. Techniques and applications in 3D bioprinting with chitosan bio-inks for drug delivery: A review. Int J Biol Macromol. 2024 Oct;278(Pt 4):134752. doi: 10.1016/j.ijbiomac.2024.134752

Khiari Z. Recent Developments in Bio-Ink Formulations Using Marine-Derived Biomaterials for Three-Dimensional (3D) Bioprinting. Mar Drugs. 2024 Mar 16;22(3):134. doi: 10.3390/md22030134

Wu S, et al. Advances in tissue engineering of gellan gum-based hydrogels. Carbohydr Polym. 2024 Jan 15;324:121484. doi: 10.1016/j.carbpol.2023.121484.

Tabatabaei Hosseini BS, et al. Biofabrication of Cellulose-based Hydrogels for Advanced Wound Healing: A Special Emphasis on 3D Bioprinting. Macromol Biosci. 2024 May;24(5):e2300376. doi: 10.1002/mabi.202300376

Patrocinio D, et al. Biopolymers for Tissue Engineering: Crosslinking, Printing Techniques, and Applications. Gels. 2023 Nov 10;9(11):890. doi: 10.3390/gels9110890

Sekar MP, et al. Hyaluronic Acid as Bioink and Hydrogel Scaffolds for Tissue Engineering Applications. ACS Biomater Sci Eng. 2023 Jun 12;9(6):3134-3159. doi: 10.1021/acsbiomaterials.3c00299

Malafaia AP, et al. Thiol-ene click chemistry: Enabling 3D printing of natural-based inks for biomedical applications. Biomater Adv. 2025 Feb;167:214105. doi: 10.1016/j.bioadv.2024.214105

Wei Q, et al. Modification, 3D printing process and application of sodium alginate based hydrogels in soft tissue engineering: A review. Int J Biol Macromol. 2023 Mar 31;232:123450. doi: 10.1016/j.ijbiomac.2023

D A G, Adhikari J, et al. 3D printing of bacterial cellulose for potential wound healing applications: Current trends and prospects. Int J Biol Macromol. 2024 Nov;279(Pt 2):135213. doi: 10.1016/j.ijbiomac.2024.135213

Pasquier E, et al. Polysaccharides and Structural Proteins as Components in Three-Dimensional Scaffolds for Breast Cancer Tissue Models: A Review. Bioengineering (Basel). 2023 Jun 3;10(6):682. doi: 10.3390/bioengineering10060682

Authors' response: We would like to thank the Reviewer for his / her constructive comments and suggestions. In order to adequately address the knowledge gap mentioned by the respectful Reviewer, we have rewritten the Introduction and have included the suggested articles.

1.2. One of the advantages of Stimuli-Responsive Polysaccharides for Tissue Engineering is the ability to create artificial tissues with properties that mimic those of biological objects. However, the lack of adaptability and dynamics is an obstacle. Therefore, it is of interest to review stimulus-sensitive polysaccharides for use in 3D bioprinting inks.

There are many review papers on this topic. Below is a short list of them. It is unclear why the authors did not analyze them to highlight the features of their review study. As a result of this omission, this manuscript is not comprehensive and of relevance to the field.

Boase NRB, et al. Stimuli-Responsive Polymers at the Interface with Biology. Biomacromolecules. 2024 Sep 9;25(9):5417-5436. doi: 10.1021/acs.biomac.4c00690

Kolipaka T, et al. Stimuli-responsive polysaccharide-based smart hydrogels for diabetic wound healing: Design aspects, preparation methods and regulatory perspectives. Carbohydr Polym. 2024 Jan 15;324:121537. doi: 10.1016/j.carbpol.2023.121537.

Asadi K, et al. Stimuli-responsive hydrogel based on natural polymers for breast cancer. Front Chem. 2024 Jan 18;12:1325204. doi: 10.3389/fchem.2024.1325204.

Tian B, Liu J. Smart stimuli-responsive chitosan hydrogel for drug delivery: A review. Int J Biol Macromol. 2023 Apr 30;235:123902. doi: 10.1016/j.ijbiomac.2023.123902

Psarrou M, et al. Stimuli-Responsive Polysaccharide Hydrogels and Their Composites for Wound Healing Applications. Polymers (Basel). 2023 Feb 16;15(4):986. doi: 10.3390/polym15040986

Soleimani K, et al. Stimuli-responsive natural gums-based drug delivery systems for cancer treatment. Carbohydr Polym. 2021 Feb 15;254:117422. doi: 10.1016/j.carbpol.2020.117422

Xia Y, et al. Advances in Stimuli-Responsive Chitosan Hydrogels for Drug Delivery Systems. Macromol Biosci. 2024 May;24(5):e2300399. doi: 10.1002/mabi.202300399

Ghasempour A, et al. Stimuli-responsive carrageenan-based biomaterials for biomedical applications. Int J Biol Macromol. 2024 Dec 18;291:138920. doi: 10.1016/j.ijbiomac.2024.138920.

Zhang Y, et al. Recent Advances of Stimuli-Responsive Polysaccharide Hydrogels in Delivery Systems: A Review. J Agric Food Chem. 2022 Jun 1;70(21):6300-6316. doi: 10.1021/acs.jafc.2c01080

Ganguly K, et al. Stimuli-responsive self-assembly of cellulose nanocrystals (CNCs): Structures, functions, and biomedical applications. Int J Biol Macromol. 2020 Jul 15;155:456-469. doi: 10.1016/j.ijbiomac.2020.03.171.

Khodadadi Yazdi M, et al. Polysaccharide-based electroconductive hydrogels: Structure, properties and biomedical applications. Carbohydr Polym. 2022 Feb 15;278:118998. doi: 10.1016/j.carbpol.2021.118998.

Authors' response: In comply with the Reviewer's suggestion, we have added stimuli responsive polysaccharide hydrogels to the manuscript so that readers can find out the importance of this subject.

  1. From a technological point of view, the use of polysaccharides as bioinks for 3D printing implies their special rheological properties. Unfortunately, the manuscript is completely lacking information on this topic. A brief description of the polysaccharides themselves in Section 3. Polysaccharides is fragmentary and does not provide an understanding of their physicochemical properties that determine the prospects for 3D printing and stimulus sensitivity.

Authors' response: Thank you so much for taking time and your suggestion. We added a new section related to bio inks to cover the special properties of bio inks.

  1. The Conclusion and Future Prospect section should be rewritten to include the conclusions drawn from the analysis, their significance, and future prospects. Repeated descriptions of results extracted from other studies should be removed.

The text (lines 1056-1064) is surprising and needs to be deleted. «In order to create a proper paste that could be extruded and form an adequate strand, chocolate was melted and combined with corn syrup. In order to create a gel that can control drug release and have a sufficient printing capacity, gelatin's viscosity must be adjusted by manipulating the temperature and its concentration in the combination. For the reasons listed below, alginate is another natural substance that can provide a gel with the ideal qualities for PAM printing. While sodium hyaluronate, astragalus root, and snake gourd root cannot be printed on their own, they do aid in the creation of a printable gel. Additionally, chitosan and sodium hyaluronate were used as adjuvants to coat the finished system»

Authors' response: Thank you so much we delete the text to mentioned in comment (Lines 1056-1064) then rewrite and restructured the Conclusion and Future Prospect section.

---------------------------------------------------------------------------------

Reviewer 2 Report (New Reviewer)

Comments and Suggestions for Authors

This manuscript aims to review recent advancements in 3D printing of stimuli-responsive polysaccharides for applications in tissue engineering and localized drug delivery. While the paper provides a general overview of 3D printing technologies and polysaccharides, there are significant areas where the manuscript needs improvement to meet the standards of a systematic, high-quality review paper.

Specific Comments:

1.        The manuscript lacks a critical discussion of the limitations of 3D printing, such as the defects and imperfections that arise from the layer-by-layer deposition process. These defects, including voids, incomplete fusion, surface roughness, and internal anisotropy, significantly impact the properties of printed structures specifically in drug delivery aspect. The authors should cite relevant studies (e.g., https://doi.org/10.1208/s12249-020-01771-4,  https://doi.org/10.1007/s00170-022-10062-0, 10.1016/j.ijpharm.2018.03.056 ), which extensively discuss how such imperfections affect material performance and reliability. Including this would provide a more balanced and realistic view of 3D printing's applications in biomedicine.

2.        The manuscript does not adopt a systematic approach to reviewing the literature. There is no clear explanation of how the reviewed studies were selected, nor any criteria for inclusion or exclusion. Refer to established guidelines for systematic literature reviews, such as https://doi.org/10.1016/j.mpsur.2009.07.005, to incorporate methodological rigor.

3.        The manuscript does not clearly articulate its unique contributions or novelty. It is unclear how this review advances the field beyond existing literature. The authors should explicitly highlight the novelty in the introduction and conclusion.

4.        The manuscript should propose actionable steps for addressing the limitations of 3D printing and polysaccharides in biomedical applications.

Author Response

Reviewer 2

This manuscript aims to review recent advancements in 3D printing of stimuli-responsive polysaccharides for applications in tissue engineering and localized drug delivery. While the paper provides a general overview of 3D printing technologies and polysaccharides, there are significant areas where the manuscript needs improvement to meet the standards of a systematic, high-quality review paper.

Specific Comments:

  1. The manuscript lacks a critical discussion of the limitations of 3D printing, such as the defects and imperfections that arise from the layer-by-layer deposition process. These defects, including voids, incomplete fusion, surface roughness, and internal anisotropy, significantly impact the properties of printed structures specifically in drug delivery aspect. The authors should cite relevant studies (e.g., https://doi.org/10.1208/s12249-020-01771-4, https://doi.org/10.1007/s00170-022-10062-0, 10.1016/j.ijpharm.2018.03.056 ), which extensively discuss how such imperfections affect material performance and reliability. Including this would provide a more balanced and realistic view of 3D printing's applications in biomedicine.

Authors' response: Thank you so much for your comment. We added a section to discussion with your mentioned articles.

  1. The manuscript does not adopt a systematic approach to reviewing the literature. There is no clear explanation of how the reviewed studies were selected, nor any criteria for inclusion or exclusion. Refer to established guidelines for systematic literature reviews, such as https://doi.org/10.1016/j.mpsur.2009.07.005, to incorporate methodological rigor.

Authors' response: We rearranged the manuscript as a review paper with complete headings and explanations.

  1. The manuscript does not clearly articulate its unique contributions or novelty. It is unclear how this review advances the field beyond existing literature. The authors should explicitly highlight the novelty in the introduction and conclusion.

Authors' response: Thank you for your comment we rewrite the whole introduction and conclusion to completely show the novelty.

  1. The manuscript should propose actionable steps for addressing the limitations of 3D printing and polysaccharides in biomedical applications.

Authors' response: We have addressed some limitations and in separate section and in general discusses about them and actionable steps in the manuscript.

---------------------------------------------------------------------------------

Reviewer 3 Report (New Reviewer)

Comments and Suggestions for Authors

This manuscript provides a comprehensive review of the application of 3D printing technology in biomedicine, with a particular focus on stimuli-responsive polysaccharides for tissue engineering and targeted drug delivery. The structure of the article is clear, and the logical flow is strong, offering a detailed discussion on the advantages of 3D printing technology and the potential of polysaccharides as biomaterials. Overall, the manuscript holds significant value but requires further refinement in certain areas:

1. The background section in the abstract lacks a sense of urgency. It would be beneficial to emphasize the importance of the manuscript, highlighting the specific scientific problems it addresses and their relevance.

2. The introduction includes too much general information, which detracts from the focus of the paper. It is recommended to narrow the scope to the central issues discussed in the manuscript, making it easier for the reader to grasp the key points.

3. The overall logic and structure of the introduction need to be better organized. For example, lines 59-61 only present an example, and this segmentation seems somewhat misplaced.

4. Table 1, which summarizes 3D printing technologies, seems incomplete. It would be advisable to verify and include additional relevant information. The table title also needs to be revised, as it is currently overly simplistic.

5. The 3rd section would benefit from a comparative list, which could enhance clarity and presentation.

Author Response

Reviewer 3

This manuscript provides a comprehensive review of the application of 3D printing technology in biomedicine, with a particular focus on stimuli-responsive polysaccharides for tissue engineering and targeted drug delivery. The structure of the article is clear, and the logical flow is strong, offering a detailed discussion on the advantages of 3D printing technology and the potential of polysaccharides as biomaterials. Overall, the manuscript holds significant value but requires further refinement in certain areas:

  1. The background section in the abstract lacks a sense of urgency. It would be beneficial to emphasize the importance of the manuscript, highlighting the specific scientific problems it addresses and their relevance.

Authors' response: Thank you so much for your comments. We changed the abstract to highlight importance of the manuscript.

  1. The introduction includes too much general information, which detracts from the focus of the paper. It is recommended to narrow the scope to the central issues discussed in the manuscript, making it easier for the reader to grasp the key points.

Authors' response: We rewrite the introduction to clarify issues and key points for better understanding.

  1. The overall logic and structure of the introduction need to be better organized. For example, lines 59-61 only present an example, and this segmentation seems somewhat misplaced.

Authors' response: Thank you for your comments. As suggested, we have completely rewritten the introduction.

  1. Table 1, which summarizes 3D printing technologies, seems incomplete. It would be advisable to verify and include additional relevant information. The table title also needs to be revised, as it is currently overly simplistic.

Authors' response: Thank you so much for your comment. We changed table 1 and its title completely. We try to include all the techniques and advantages and disadvantages.

  1. The 3rd section would benefit from a comparative list, which could enhance clarity and presentation.

Authors' response: Table 3 is a comparative list of printed scaffolds made of polysaccharides.

---------------------------------------------------------------------------------

Reviewer 4 Report (New Reviewer)

Comments and Suggestions for Authors

In the manuscript entitled “3D Printing of Stimuli-Responsive Polysaccharides for Tissue Engineering and Localized Drug Delivery: A Review”, authors explore various 3D printing technologies and the application of stimuli-responsive polysaccharides in 3D printing. However, the article lacks novelty, suffers from disorganized structural logic between paragraphs, omits essential illustrative figures, and fails to establish a strong connection between its conclusion and the core topic. The specific issues identified are as follows:

1. In “Introduction” section, the introduction is overly lengthy and should concisely present the topic and the reasons for its discussion. The logical relationships between paragraphs are unclear, making it difficult for readers to discern the author’s objectives.

2. In “3D Printing Technologies section”, the rationale behind the selection of 3D printing technologies is not adequately explained. The author should prioritize techniques relevant to bioprinting rather than providing a broad, superficial overview of various 3D printing technologies. The content describing these technologies is outdated and lacks depth, failing to align with the central theme of the article.

3. In “Polysaccharides” section, the discussion on polysaccharides is overly simplistic. While briefly introducing the properties of polysaccharides is acceptable, the emphasis should be on their applications in 3D bioprinting.

4.  In “3D Printing Hydrogels for Tissue Engineering and drug delivery” section, this section should be the article’s core focus. A clear and logical structure, potentially using subheadings to classify discussions (e.g., polysaccharide 3D printing under different stimuli conditions or targeting specific tissues or diseases), is necessary. Current content lacks coherence, and the paragraph structure is disorganized, making it challenging for readers to extract meaningful information.

5. The article would benefit from summary figures to assist readers in understanding the content and its connections.

6. Several highlighted sections are present throughout the article, but their purpose is unclear. The author should clarify what these annotations aim to convey.

7. In “Discussion” section: The discussion fails to adequately summarize and analyze stimuli-responsive polysaccharides, leading to a weak connection with the article’s central theme.

Author Response

Reviewer 4

In the manuscript entitled “3D Printing of Stimuli-Responsive Polysaccharides for Tissue Engineering and Localized Drug Delivery: A Review”, authors explore various 3D printing technologies and the application of stimuli-responsive polysaccharides in 3D printing. However, the article lacks novelty, suffers from disorganized structural logic between paragraphs, omits essential illustrative figures, and fails to establish a strong connection between its conclusion and the core topic. The specific issues identified are as follows:

  1. In “Introduction” section, the introduction is overly lengthy and should concisely present the topic and the reasons for its discussion. The logical relationships between paragraphs are unclear, making it difficult for readers to discern the author’s objectives.

Authors' response: We restructure the introduction section to clarify the present topic and making it easy for readers.

  1. In “3D Printing Technologies section”, the rationale behind the selection of 3D printing technologies is not adequately explained. The author should prioritize techniques relevant to bioprinting rather than providing a broad, superficial overview of various 3D printing technologies. The content describing these technologies is outdated and lacks depth, failing to align with the central theme of the article.

Authors' response: In order to providing better understanding we added all 3D printing technologies to manuscript.

  1. In “Polysaccharides” section, the discussion on polysaccharides is overly simplistic. While briefly introducing the properties of polysaccharides is acceptable, the emphasis should be on their applications in 3D bioprinting.

Authors' response: We added all polysaccharides types to the manuscript and try to show their properties and applications in 3D bioprinting.

  1. In “3D Printing Hydrogels for Tissue Engineering and drug delivery” section, this section should be the article’s core focus. A clear and logical structure, potentially using subheadings to classify discussions (e.g., polysaccharide 3D printing under different stimuli conditions or targeting specific tissues or diseases), is necessary. Current content lacks coherence, and the paragraph structure is disorganized, making it challenging for readers to extract meaningful information.

Authors' response: We rewrite this section and cut it to 3 sub sections that most used polysaccharide name with recent advances.

  1. The article would benefit from summary figures to assist readers in understanding the content and its connections.

Authors' response: Thank you so much for your comment and suggestion we added two new figures for 2 different sections.

  1. Several highlighted sections are present throughout the article, but their purpose is unclear. The author should clarify what these annotations aim to convey.

Authors' response: Highlighted sections are presenting the latest changes after revision as requested by the Journal Editorial team.

  1. In “Discussion” section: The discussion fails to adequately summarize and analyze stimuli-responsive polysaccharides, leading to a weak connection with the article’s central theme.

Authors' response: We would like to express our gratitude to the Reviewers for their valuable and constructive suggestions. We completely changed the Discussion section to summarize stimuli responsive polysaccharides and bio inks and challenges.

-------------------------------------------

Round 2

Reviewer 1 Report (Previous Reviewer 2)

Comments and Suggestions for Authors

see attached file

Author Response

The authors have substantially supplemented the review with information. However, the general idea of the review and its conclusions remain rather superficial. The three lines of reasoning barely intersect with each other, so it is not clear why they are combined in one article. The first is a brief listing of polysaccharides, the second gives some examples of stimulus-responsive gels constructed with them, and the third describes experimental papers using polysaccharides in 3D printing, without analyzing the stimulus-responsiveness of the printed structures. This presentation of the material is completely inconsistent with the title of the article. Therefore, it is strongly recommended that the authors change the title of the article. In addition to the above disadvantage, the manuscript describes stimulus-responsive gels, which is correct, and not stimulus-responsive polysaccharides, which is rather an uncommon term. In addition, the work completely lacks an analysis of "Localized Drug Delivery" of anything from any 3D printed gel.

Answer: We changed our article title here is the new title “3D Printing of Hydrogels Polysaccharides for Biomedical Ap-plications: A Review”.

It is unclear why the introduction contains information about 3D printing methods that are not used to print stimulus-sensitive polysaccharide hydrogel structures.

It is strange that alginate and its gels are described so briefly, although this is the most common and studied material for bioinks.

Why is section 8. Limitations limited to discussions on cardiac tissue engineering issues?

Answer: Other reviewer wanted a separated section for Limitations as a most important tissue we discuss shortly about cardiac tissue.

Paragraph Lines 1670-1684 are not appropriate in this section.

Answer: Thank you. It was a mistake in a last revision. 

Reviewer 2 Report (New Reviewer)

Comments and Suggestions for Authors

the authors addressed all the comments.

Author Response

the authors addressed all the comments.

Answer: We compeletly changed the structure of abstract, introduction and list of contents they are changed into red texts and highlighted yellow. Lines 1618 to 1644 are the answers to comment 1.

Reviewer 4 Report (New Reviewer)

Comments and Suggestions for Authors

I am pleased to see that the authors have made significant efforts to revise the manuscript based on the suggestions provided. The quality of the manuscript has improved to some extent. However, there are still several issues that require careful consideration:

  1. The keywords in the title include "3D printing," "stimuli-responsive," "polysaccharides," "tissue engineering," and "localized drug delivery." The scope covered in the title is too broad, making it difficult for the authors to maintain a focused discussion throughout the review. For example, while the manuscript provides a detailed introduction to various 3D printing technologies, many of them are not applicable to bioprinting. Additionally, the discussion on stimuli-responsive hydrogel has little direct connection to 3D printing, and the section on polysaccharide-based 3D printing does not strongly relate to stimuli-responsive. As a result, the review appears fragmented rather than centered around a coherent theme.

  2. In the discussion section, both the limitations and future perspectives primarily focus on 3D printing technologies in general, while the main content of the manuscript is about polysaccharide-based 3D printing. This broad and somewhat unfocused discussion lacks a strong connection to the core theme of the manuscript. Despite the extensive additions made during revision, the authors should ensure that all content is closely related to the central theme. A review is not merely a collection of relevant information—it should maintain internal coherence and logical connections between different sections. The main issue with the current version of the manuscript is that while it covers a lot of content, the relationship between the sections and the main theme is unclear, leading to a somewhat disorganized structure. The authors should carefully reconsider the central focus of this review and establish a clear logical framework for different sections, rather than simply expanding the content.

Comments on the Quality of English Language

The language is understandable, however, some expressions are not entirely accurate. Refining word choices and sentence structures would enhance clarity and precision.

Author Response

I am pleased to see that the authors have made significant efforts to revise the manuscript based on the suggestions provided. The quality of the manuscript has improved to some extent. However, there are still several issues that require careful consideration:

    1. The keywords in the title include "3D printing," "stimuli-responsive," "polysaccharides," "tissue engineering," and "localized drug delivery." The scope covered in the title is too broad, making it difficult for the authors to maintain a focused discussion throughout the review. For example, while the manuscript provides a detailed introduction to various 3D printing technologies, many of them are not applicable to bioprinting. Additionally, the discussion on stimuli-responsive hydrogel has little direct connection to 3D printing, and the section on polysaccharide-based 3D printing does not strongly relate to stimuli-responsive. As a result, the review appears fragmented rather than centered around a coherent theme.

Answer: We changed article name and keywords stimuli-responsive and localized drug delivery deleted so it will focus on 3d printing hydrogels polysaccharides and biomedical applications.

    2. In the discussion section, both the limitations and future perspectives primarily focus on 3D printing technologies in general, while the main content of the manuscript is about polysaccharide-based 3D printing. This broad and somewhat unfocused discussion lacks a strong connection to the core theme of the manuscript. Despite the extensive additions made during revision, the authors should ensure that all content is closely related to the central theme. A review is not merely a collection of relevant information—it should maintain internal coherence and logical connections between different sections. The main issue with the current version of the manuscript is that while it covers a lot of content, the relationship between the sections and the main theme is unclear, leading to a somewhat disorganized structure. The authors should carefully reconsider the central focus of this review and establish a clear logical framework for different sections, rather than simply expanding the content.

Answer: Thank you so much. We rewrite future perspective but during several revision other reviewer wants us to added some extra information so we just trying to cover all the reviewer comments. However; We generally focus on 3d printing and recent 3D polysaccharide advances.

Round 3

Reviewer 1 Report (Previous Reviewer 2)

Comments and Suggestions for Authors

The new title “3D Printing of Hydrogels Polysaccharides for Biomedical Applications: A Review” made the material presented in the manuscript more logical. Now the manuscript can be recommended for publication. 

Reviewer 4 Report (New Reviewer)

Comments and Suggestions for Authors

After making the revisions, the quality of the manuscript has been significantly improved. I agree to its publication in the journal.

This manuscript is a resubmission of an earlier submission. The following is a list of the peer review reports and author responses from that submission.

Round 1

Reviewer 1 Report

Comments and Suggestions for Authors

Dear colleagues!

I have carefully studied your review "3D Printing of Stimuli-Responsive Polysaccharides for Tissue Engineering and Localized Drug Delivery: A Review". You analyzed a large volume of literature while working on the review, and therefore it is interesting. I was glad to read it. However, I have a number of questions.

509-510. Pullulan methacrylate and PEGDA were combined by Giustina et al. to enable multiscale light-assisted 3D printing methods, including stereolithography and two-photon lithography [136].

In other places, when there are references, you do not indicate the authors of the article. Why did you indicate this time?

Lines 537-552.

 What polysaccharide did you write about the use of? It is not indicated from which polysaccharides the framework was printed. Only the gelatin protein is mentioned.

558-559 All four scaffolds were shown to be significantly hydrophilic and could be gradually degraded.

You have not mentioned 4 scaffolds before.

605-619

How did the dual drug scaffolds inhibit biofilm formation and temporarily stop S. Aureus growth? If the review focuses on polysaccharides, it is necessary to specify the role they play in each of the described cases.

620-627 . Fe3O4 and cellulose nanofibers (CNF) in combination with polyhydroxybutyrate/poly(ε-caprolactone) (PHB/PCL) blends as functional particles and reinforcements created a novel composite with remarkable shape memory and tensile strength. We evaluated the effects of CNF addition and printing conditions on the mechanical properties and magnetically responsive shape memory of the 3D printed parts. Fe3O4 was added to the PHB/PCL combination to increase tensile strength and decrease elongation at break. CNFs significantly affected the strengthening and hardening of the PHB/PCL blend with 10% Fe3O4 at a loading rate of 0.5 wt%.

Is this data from your study? There should be a citation.

1096 Contrary to popular belief, hydrogels composed solely of alginate are completely degradable.

Alginate hydrogels can dissolve, but they do not degrade. This is a misnomer

1143-1145 Although we considered a variety of biomaterials from both terrestrial and marine sources in this analysis, it should be emphasized that alginate and nanocellulose have emerged as the leading contenders for use in 3D printing.

According to PubMed (a free search engine for biomedical research) created by the National Center for Biotechnology Information (NCBI), when searching for the keywords 3D bioprinting and the name of the polysaccharide, the following number of articles were found:

3D printing hyaluronic acid -324 links

3D printing nanocellulose -132 links

3d printing alginate 1,165 results

3D printing chitosan -440

3D printing cellulose -763

3D printing gum- 164.

Nanocellulose showed the fewest links to articles in this list.

Chapter 3 describes many different polysaccharides, but some polysaccharides that are very important for 3D printing (alginate, hyaluronic acid) are described very briefly, unlike gums. Chitin and cellulose are not used for bioprinting, but they also receive a lot of attention. A more equivalent description of polysaccharides is needed. In Chapter 3, you paid a lot of attention to gums. But in Chapter 4, you don't write about them. But gellan, xanthan and other gums are often used in 3D printing.

In Chapter 4, the results are presented chaotically. It is necessary to structure the material. This text is a set of separate facts that are not related to each other.

The works of outstanding researchers from Europe, Russia and America are not mentioned: Kevin Shakeshaff, Lothar Koch, Gabor Forgacs, Utkan Demirci, Günter Tovar, Vladimir Mironov and others.

Not everywhere are there any abbreviations.

Best regards

Reviewer 2 Report

Comments and Suggestions for Authors

see attached file

Comments on the Quality of English Language

The English could be improved to more clearly express the research.

Reviewer 3 Report

Comments and Suggestions for Authors The authors summarized polysaccharides for tissue engineering and drug delievery, including 3D printing. The review is well-organized, it is quite extensive. The topic is of interest. The references are appropriate. I have only minimum comments: I do not see connection of carboxymethyl cellulose, bacterial cellulose , chitin, alginate, pectin and hyaluronic acid with 3D printing.  Are there no more references from year 2024? Reference 139 is not complete.